# BFS-Prover-V2: Scaling up Multi-Turn Off-Policy RL and Multi-Agent Tree Search for LLM Step-Provers

## Abstract

The integration of Large Language Models (LLMs) with automated theorem proving has shown immense promise, yet is constrained by challenges in scaling up both training-time reinforcement learning (RL) and inference-time compute. This paper introduces BFS-Prover-V2, a step-level theorem proving system designed to address this dual scaling problem. We present two primary innovations. The first is a novel multi-turn off-policy RL framework for continually improving the performance of the LLM step-prover at training time. This framework, inspired by the principles of AlphaZero, utilizes a multi-stage expert iteration pipeline featuring adaptive tactic-level data filtering and periodic retraining to surmount the performance plateaus that typically curtail long-term RL in LLM-based agents. The second innovation is a planner-enhanced multi-agent system that scales reasoning capabilities at inference time. This architecture employs a general reasoning model as a high-level planner to iteratively decompose complex theorems into a sequence of simpler subgoals. This hierarchical approach substantially reduces the search space, enabling a team of parallel prover agents to collaborate efficiently by leveraging a shared proof cache. We demonstrate that this dual approach to scaling yields state-of-the-art results on established formal mathematics benchmarks. BFS-Prover-V2 achieves 95.08% and 41.4% on the miniF2F and ProofNet test sets respectively. While demonstrated in the domain of formal mathematics, the RL and inference techniques presented in this work are of broader interest and may be applied to other domains requiring long-horizon multi-turn reasoning and complex search.

## 1 Introduction

Automated Theorem Proving (ATP), a subfield of mathematical logic and automated reasoning, represents one of the foundational ambitions of computer science (Bibel et al., 1993). The contemporary landscape of formal mathematics is increasingly dominated by interactive theorem provers (ITPs) or proof assistants. These systems, such as Coq, Isabelle, and Lean, require a human user to guide the proof process, but they automate significant deductive tasks and, most importantly, provide a machine-checkable guarantee of correctness Geuvers (2009). Among these, the Lean4 programming language (Moura & Ullrich, 2021) has emerged as a particularly vibrant ecosystem. A key factor in its success is Mathlib (Blokpoel, 2024), a vast and comprehensive, community-driven library of formalized mathematics. Spanning over a million lines of code, mathlib covers extensive areas of algebra, analysis, topology, and more, providing a rich foundation for both advanced mathematical research and the development of verified systems.

The rise of Lean4 has coincided with the explosion in the capabilities of LLMs (OpenAI, 2023; Comanici et al., 2025; Seed et al., 2025), opening a new frontier in neuro-symbolic AI systems. The goal here is to integrate the intuitive yet powerful generation and search capabilities of LLMs with the absolute logical verification of formal systems. This research direction centers on a key feedback loop: an LLM proposes intuitive proof steps, the Lean compiler provides rigorous verification, and RL Sutton (2018) uses that verification to continuously improve the LLM's reasoning abilities (Yang et al., 2024b; Xin et al., 2024a; Polu et al., 2022; Han et al., 2021; Lample et al., 2022).

## 1.1 MOTIVATION: A DUALITY OF SCALING CHALLENGES IN REASONING

The development of high-performance formal math provers, or any other reasoning agents, is contingent upon solving two fundamental and deeply interconnected scaling challenges.

**Training-time scaling.** This refers to the techniques required to continuously enhance a model's foundational capabilities and tactical intuitions via training. A common and significant obstacle in applying RL to LLMs is the phenomenon of performance plateaus: after an initial phase of rapid improvement, models often stagnate, with their capabilities ceasing to grow despite continued training (Liu et al., 2025; Team et al., 2025; Yu et al., 2025; Yue et al., 2025; Guo et al., 2025; Seed et al., 2025; Xin et al., 2024a;b). Overcoming this limitation requires carefully designed algorithms that can sustain learning over extended periods, enabling the model to transition from mastering simple problems to tackling increasingly complex theorems.

**Inference-time scaling.** This addresses the method of using a trained model to solve previously unseen theorems. The primary bottleneck here is inefficient exploration in a vast search space. Specifically, pure tree search is often constrained by a lack of global planning ability and a search space that grows exponentially with proof depth. More generally, current inference paradigms suffer from the fact that search trajectories are independent, meaning that insights gained in one proof attempt are not shared with others. The challenge, therefore, is to design an inference architecture that incorporates planning and collaborative search to more effectively allocate computational resources (Baba et al., 2025; Zhou et al., 2025; Chen et al., 2025; Jiang et al., 2022; Cao et al., 2025).

## 1.2 OUR CONTRIBUTIONS

This paper presents `BFS-Prover-V2`, a comprehensive training and inference system for neural theorem proving in Lean4 that introduces novel solutions to the above scaling challenges. The primary contributions of this work are as follows:

**Novel RL Scaling Techniques at Training:** We develop a distillation-free multi-stage expert-iteration framework (Silver et al., 2018; Anthony et al., 2017), a form of off-policy RL, tailored for the domain of formal theorem proving. To sustain learning and overcome performance plateaus, we introduce a suite of specialized techniques within the RL pipeline. These include an adaptive, perplexity-based data filtering strategy at the tactic level, which creates an automated curriculum for the agent, and a periodic retraining mechanism that acts as a "soft reset" to escape local optima in the model parameter space and increase model scaling potential.

**Multi-Agent Tree Search System at Inference:** For inference-time scaling, we introduce a hierarchical reasoning architecture. A general-purpose reasoning model, termed the planner, provides global planning ability by iteratively decomposing complex theorems/goals into a sequence of more manageable subgoals. These subgoals serve as tree search "checkpoints" with successful proof trajectories stored in a shared cache. This dramatically reduces search complexity by converting the total computational effort from a product of individual subgoal complexities to their sum.

**State-of-the-Art Empirical Results:** We validate the effectiveness and generalizability of our dual scaling approach on established benchmarks. In particular, `BFS-Prover-V2` achieves 95.08% on the miniF2F test set, largely surpassing previous LLM step-provers (Wu et al., 2024a; Xin et al., 2025) and performing on par with best whole proof generation models (Ren et al., 2025; Lin et al., 2025b; Wang et al., 2025). On ProofNet-test, it achieves 41.4%, setting a new state-of-the-art and demonstrating robust generalization across distributions.

## 2 THE BFS-PROVER-V2 SYSTEM

This section details the two core components of `BFS-Prover-V2`: (i) a training pipeline, grounded in a Markov Decision Process (MDP) Puterman (1990) and scaled via adaptive filtering and periodic retraining; and (ii) an inference engine, which uses a planner-enhanced multi-agent search for hierarchical reasoning. These components build upon the foundation of `BFS-Prover-V1` Xin et al. (2025) to specifically address the dual challenges of scaling at both training and inference time. We provide visual overviews of these components in Fig. 1 and 3, with practical implementation parameters and ablations detailed in Section 3.

## 2.1 A STEP-LEVEL FORMULATION: THEOREM PROVING AS A MARKOV DECISION PROCESS

We formulate proof search in Lean4 tactic mode as a Markov Decision Process (MDP), where an LLM-based prover (agent) interacts with the Lean proof checker (environment) to construct formal proofs. This formulation naturally captures the sequential, stateful nature of tactic-based formal theorem proving. Formally, we define the MDP tuple $\mathcal{M} = (S, A, P, R)$ as follows:

- **State Space** $S$: Each state $s \in S$ corresponds to a tactic state returned by the Lean compiler, comprising the current hypotheses (known facts) and target goals to be proven.
- **Action Space** $A$: An action $a \in A$ is a tactic string generated by the prover. Each tactic encodes a proof step, such as theorem application, term rewriting, or goal decomposition, that instructs the compiler to perform a specific deductive transformation.
- **Transition** $P$: The transition function $P(s' \mid s, a)$ is deterministically executed by the Lean proof checker. Given state $s$ and action $a$, the compiler either produces a successor state $s'$ if $a$ is applicable, or returns an error, resulting in a terminal failure state.
- **Reward Function** $R$: We employ sparse rewards where $R(s, a) = 1$ if and only if the state-action pair $(s, a)$ lies on a trajectory culminating in a successful proof trajectory. Otherwise, $R(s, a) = 0$.

This tactic-level stepwise interactive formulation fundamentally differs from whole-proof generation approaches (Ren et al., 2025; Lin et al., 2025b; Wang et al., 2025), which frame theorem proving as a one-shot code generation task from problem statements to complete proof scripts. While simpler, such approaches cannot adapt to intermediate proof states and lack integration with interactive theorem proving workflows (Welleck & Saha, 2023; Song et al., 2024). Our MDP-based approach, by design, trains an agent that functions as a genuine Lean copilot, capable of suggesting appropriate tactics at each step in the proof process Yang et al. (2024c).

## 2.2 SCALING UP TRAINING: MULTI-STAGE EXPERT ITERATION

The core training loop of `BFS-Prover-V2` is an expert iteration pipeline, which may be viewed as a variant of the reinforcement learning algorithm used in AlphaZero Anthony et al. (2017); Silver et al. (2018). This approach enables the system to learn and improve its theorem-proving capabilities from its own experience Silver & Sutton (2025). The process, illustrated in the inner loop of Fig. 1, includes two major alternating phases: proof generation and model refinement.

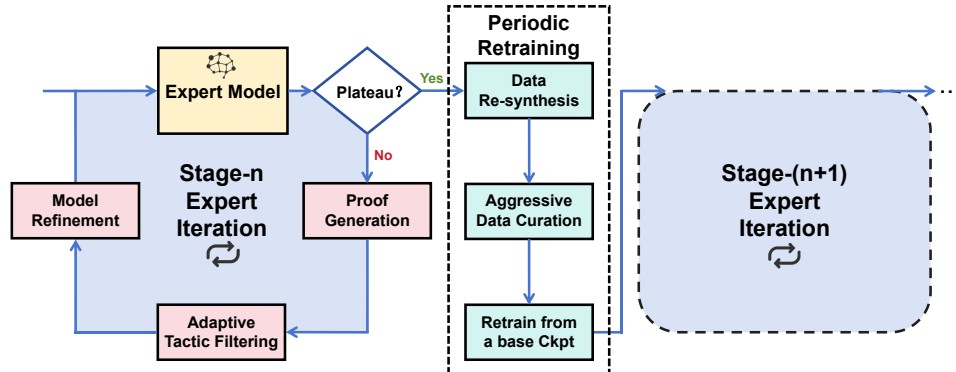

Figure 1: Overview of the training-time scaling up architecture. The process begins with a current expert model. The system then evaluates the model's performance to check for a plateau. If performance is improving, the model enters an inner **expert iteration loop**. Conversely, if performance has plateaued, the system triggers the outer **retraining loop**. The retraining loop yields an improved expert model serving as the starting point for the next cycle.

**Phase 1: Proof Generation:** The current best version of the LLM prover, referred to as the expert, is tasked with solving a large corpus of mathematical problems. The expert model is coupled with the best-first tree search (BFS) algorithm used in `BFS-Prover-V1` Xin et al. (2025) to explore the

vast space of possible proof trajectories. Each successful proof found during this phase constitutes a trajectory of (state, tactic) pairs. Across a single round of expert iteration, the system performs approximately $10^7$ tree searches, generating a massive synthetic dataset.

**Phase 2: Model Refinement:** The state-tactic pairs from the successful proof trajectories generated in the first phase are used to update the model's parameters. The updated model then becomes the new "expert" for the next round of iteration.

A central challenge in scaling the expert iteration pipeline or RL in general is managing the vast quantity and variable quality of the generated trajectories. Naively training on every successful trajectory quickly leads to diminishing returns, performance stagnation, and mode collapse Liu et al. (2025); Sutton (2018); Xin et al. (2025). To sustain improvement over many iterations, we introduce two key algorithmic innovations: a dynamic, fine-grained data filtering strategy and a periodic full-model retraining process. These techniques work in concert to form an automated curriculum that continuously improves the agent's capability in a long horizon. The overall architecture of this pipeline is illustrated in Fig. 1, and we detail each of these innovations in the following subsections.

### 2.2.1 ADAPTIVE TACTIC FILTERING: LEARNING FROM THE "JUST RIGHT" DATA

Instead of relying on coarse, problem-level filtering Yu et al. (2025); Team et al. (2025), which often uses static metrics of difficulty, we adopt a more fine-grained approach at the tactic level. This strategy is guided by the empirical observation that the perplexity (negative log-probability) of tactics generated by the LLM roughly follows a Gaussian distribution. The distribution, shown in Fig. 2, can be divided into three distinct regions, each with different implications for learning:

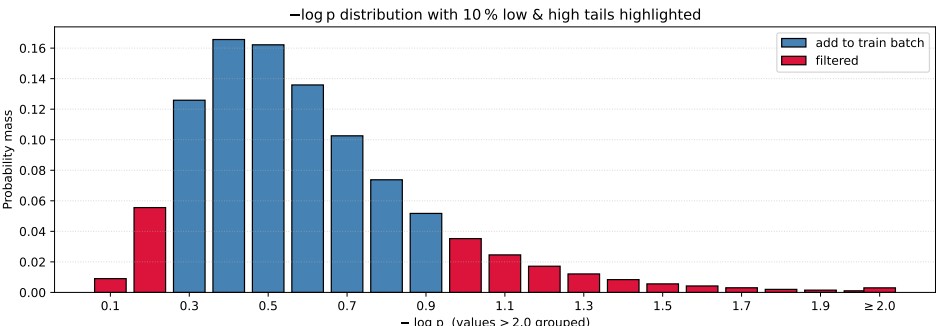

Figure 2: Tactic-Level Data Filtering Based on the Perplexity Distribution. This histogram shows the probability distribution of tactic perplexity (represented as normalized negative log-probability) from a single round of expert iteration. We filter out the low- and high-perplexity tails, shown in red.

- **The Low-Perplexity Tail:** This region corresponds to tactics for which the model has very high confidence. These are typically simple, "obvious" steps, such as basic simplification or applying a clear-cut hypothesis. Including these examples in the training batch offers no new learning signal; it merely reinforces what the model already knows well and can contribute to overfitting and a reduction in exploratory capacity.

- **The High-Perplexity Tail:** This region represents tactics that the model finds highly surprising. Our case studies reveal that these are often not instances of brilliant, non-obvious reasoning. Instead, they frequently correspond to noisy or suboptimal choices, such as using a powerful, general-purpose tactic with many unnecessary parameters on a simple problem where a more direct tactic would suffice. These "fancy" operations can be detrimental to training, as they may teach the model to generate overly complex or irrelevant tactics, leading to hallucinations and degrading its core reasoning ability.

- **The Central Distribution:** This is the "goldilocks" zone. The tactics in this region are neither too easy nor too noisy. They represent steps that are challenging for the model but still within its grasp. By selectively training only on the data from this central part of the distribution, we ensure that the model is constantly learning at the edge of its capabilities.

This adaptive filtering mechanism functions as a fully automated form of curriculum learning. It does not rely on any external/predefined metric of difficulty. Instead, it uses the model's own uncertainty (as measured by perplexity) as a dynamic signal of what constitutes valuable training data at its current stage of development. This ensures a smooth and stable evolution of the model's internal policy distribution throughout training, enabling sustained growth in performance. To further illustrate this filtering strategy, we present real-world examples of tactics falling into these three categories.

**1. Low-Perplexity Tail**

*Trivial tactics (**Discard**)*

```
x:  ℝ
h₀:  x ∈ Set.Icc (π/2)(3π/2)
h₁:  0 ≤ cos x
⊢ π/2 ≤ x ∧ x ≤ 3π/2
```

```
exact h₀
```

**2. High-Perplexity Tail**

*Noisy/hallucinated tactics (**Discard**)*

```
p:  ℝ
h₀:  0 < p
hp:  ¬p = 0
h₂:  (800+5*p)*(7*p)=800*10*p
⊢ 7*p = 48*10
```

```
nlinarith [mul_pos h₀ (show 0
    < p by linarith), mul_pos
     h₀ (by linarith: 0 < p
    /2), mul_pos h₀ (by
    linarith : 0 < p)]
```

**3. Mid-Perplexity Zone**

*Informative tactics (**Keep**)*

```
x:  ℝ
hx:  x = 250000
⊢ x = 2.5 * 10^5
```

```
norm_num [hx]
```

### 2.2.2 PERIODIC RETRAINING: A "SOFT RESET" TO ESCAPE LOCAL OPTIMA

Even with adaptive filtering, performance eventually plateaus as the prover's tactic preferences become increasingly rigid, causing it to overfit to a narrow set of proof patterns and settle into a local optimum. It becomes very good at solving problems in particular ways, but loses the ability to discover novel approaches required for harder or new problems. To escape local optima and reinvigorate the learning process, we introduce a periodic "soft reset" procedure. This constitutes a multi-stage expert-iteration process designed to increase the model's entropy and reset its exploratory potential without losing the competence it has already gained. The procedure is as follows:

1. **Re-synthesis and De-noise:** We re-run the current expert prover on the full historical problem set to regenerate all proofs using its improved policy. Since the prover at this stage is substantially stronger than the checkpoints that produced the earlier trajectories, the newly synthesized proofs are typically shorter, structurally cleaner, and contain fewer spurious steps. This pass serves as a model-driven denoising phase: it replaces outdated trajectories with higher-quality ones and removes redundant or circuitous reasoning patterns that accumulated during earlier, less capable iterations.

2. **Aggressive Data Curation:** The new, higher-quality proofs generated in the data re-synthesis phase are then subjected to an aggressive version of the tactic-level perplexity filtering described above. A much larger portion of the data is discarded, retaining only the most crucial and informative tactic steps.

3. **Retrain from a base Checkpoint:** The existing training data is completely replaced by this new, highly curated, and compact dataset. A fresh model instance is then initialized from the base checkpoint and trained from scratch on this refined data.

The resulting model initially exhibits a temporary drop in performance. This is expected, as it has been trained on a smaller, more focused dataset and has forgotten some of its previous stylistic biases. However, this new model possesses a significantly higher exploratory potential. When it is reintroduced into the expert iteration loop, its increased capacity for exploration allows it to discover novel proof strategies that were inaccessible to the previous, over-specialized model. Consequently, ts performance rapidly recovers and then surpasses the previous peak.

### 2.3 SCALING UP INFERENCE: PLANNER-ENHANCED MULTI-AGENT SEARCH

Many complex proofs in mathematics are not found through a linear sequence of simple deductions but rather through a hierarchical process of identifying and proving crucial intermediate results. Likewise, we introduce a hierarchical inference architecture, shown in Fig. 3, that divides the labor of theorem proving between two distinct agents: a high-level planner and a low-level prover.

**Planner:** This is a general-purpose reasoning LLM tasked with goal decomposition. Given the current theorem statement and proof progress, its role is not to generate a specific tactic but to propose

a high-level plan that includes a series of intermediate subgoals. By formulating these subgoals, the planner effectively transforms a single, monolithic, and potentially intractable search problem into a structured sequence of smaller, more manageable ones. This decomposition substantially reduces the dimensionality of the search space that the prover must explore.

**Prover:** This is the specialized LLM tactic generator trained via our multi-stage expert iteration pipeline described in Section 2.2. It receives one subgoal at a time from the planner and uses its learned policy, in conjunction with Best-first tree search algorithm (Xin et al., 2025), to find a formal proof for that specific subgoal.

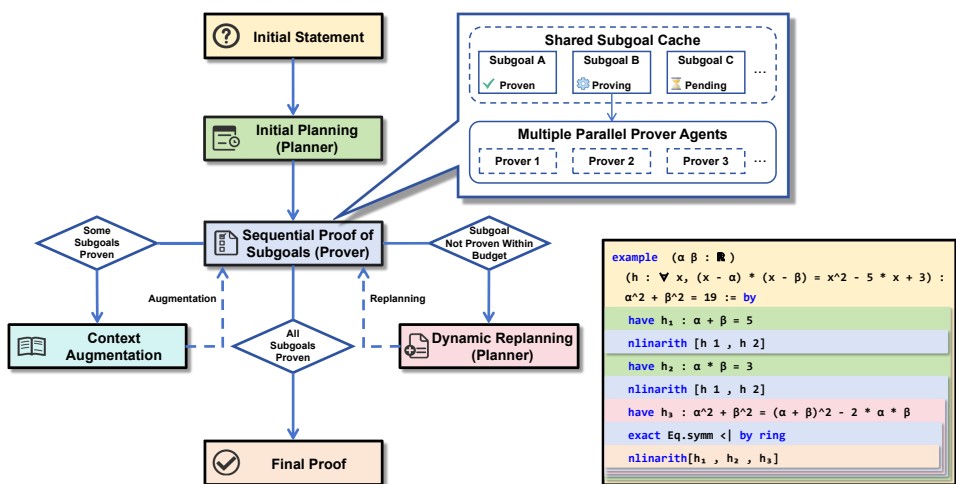

Figure 3: Overview of the multi-agent tree search architecture. The **Planner** agent decomposes the main theorem into a sequence of simpler subgoals, which are managed in a **Shared Subgoal Cache** and solved in parallel by multiple **Prover** agents. Successfully proven subgoals augment the main proof's context, while failures trigger a **Dynamic Replanning** loop. The inset provides an example, demonstrating how proving intermediate lemmas ($h_1$, $h_2$, $h_3$) facilitates the proof of the final goal.

This division of labor mirrors the cognitive workflow of a human mathematician, who might first sketch out the high-level structure of a proof by identifying key lemmas (the planner's role) and then proceed to work out the detailed, step-by-step deductions for each lemma (the prover's role). This hierarchical structure is a powerful architectural pattern for tackling complex reasoning tasks.

### 2.3.1 OPERATIONAL MECHANICS OF PLANNER-GUIDED SEARCH

As shown in Fig. 3, the interaction between the planner and the prover system unfolds in a dynamic loop, allowing the plan to be revised as proof search progresses:

1. **Initial Planning:** At the start of a proof attempt, the planner is queried with the main theorem statement. It returns a list of proposed subgoals, formatted as Lean `have` statements.

2. **Sequential Proof of Subgoals:** The prover system addresses the subgoals one by one. It takes the first subgoal in the queue and initiates tree search to find its proof.

3. **Context Augmentation:** When a subgoal is solved, its statement is incorporated into the proof context and becomes available as a hypothesis for all remaining subgoals and the main theorem itself.

4. **Dynamic Replanning:** If the prover exhausts its search budget on a subgoal, which may occur either because the subgoal is intrinsically difficult or because the planner previously produced an incorrect subgoal, the system does not terminate. Instead, the planner is invoked again with an augmented input containing the theorem statement and all previously proven subgoals. The planner then generates a revised decomposition that typically corrects, refines, or further subdivides the subgoal where search stalled.

This dynamic and iterative loop between planning and proving makes the `BFS-Prover-V2` system resilient to getting stuck, effectively scaling its inference-time capabilities to tackle complex theorems that would be intractable for a monolithic tree search.

### 2.3.2 MULTI-AGENT COLLABORATION VIA FOCUSED PARALLELISM AND SHARED SUBGOAL CACHE

To further scale inference-time compute, we deploy the planner–prover architecture in a multi-agent setting in which several prover instances run in parallel. These agents jointly execute the subgoal sequence proposed by the planner, coordinated through two design principles: focused parallelism and a shared subgoal cache.

**Focused parallelism:** Rather than distributing different subgoals in parallel across agents, all prover instances allocate their full search budget to a single active subgoal before the system advances to the next. This sequential execution concentrates search effort on difficult reasoning bottlenecks where only a subset of provers would need substantially more time to progress and avoids wasted compute on downstream subgoals that would be invalidated if an earlier step fails and triggers a replan.

**Shared Subgoal Cache:** This cache is the central communication and state-tracking mechanism, shared across all parallel prover instances. It stores the full sequence of subgoals generated by the planner, tracks the real-time status of each subgoal (e.g., pending, proving, proven), and records the proof for any solved subgoal.

This architecture creates a cooperative sprint for each lemma in the plan. When a new subgoal is designated as the active target, all prover agents begin independent tree searches for that single subgoal in parallel. As soon as the first agent finds a valid proof, it writes the result to the shared cache. The subgoal cache signals all other agents to terminate their search, preventing redundant computation. The entire group of agents then proceeds to the next subgoal in the sequence.

## 3 EXPERIMENTS

### 3.1 PRACTICAL IMPLEMENTATION

**Base model and Data:** Our prover is built upon Qwen2.5-Math-7B and Qwen2.5-32B Yang et al. (2024a), which serve as the base for our policy optimization. The multi-stage expert iteration process was initialized from the checkpoint `BFS-Prover-V1` Xin et al. (2025). To construct a large-scale training corpus, we autoformalized the NuminaMath-CoT and NuminaMath-1.5 datasets Li et al. (2024a) by prompting general-purpose LLMs, augmented with Lean4 compiler feedback. Together with Goedel-Pset-V1 Lin et al. (2025a), this yields roughly 3 million formal statements Wu et al. (2024b); Ying et al. (2024); Blokpoel (2024). Prompts used for autoformalization can be found in Section C.1. All experiments are conducted in Lean v4.10.0 with LeanDojo Yang et al. (2024c).

**Training setup:** After each expert iteration round, we refine the policy LLM using one of two supervised fine-tuning (SFT) strategies, selected based on the outcome of the round. In the inner expert-iteration loop, we fine-tune the current best checkpoint for one epoch with cosine learning rate decay from $5 \times 10^{-6}$ to $1 \times 10^{-7}$. When periodic retraining is triggered (Section 2.2), we train for three epochs with a larger learning rate that decays from $2 \times 10^{-5}$ to $1 \times 10^{-6}$. Both strategies use a global batch size of 1024.

**Inference configuration:** Our inference pipeline combines a low-level prover with a high-level planner, as detailed in Section 2.3. Prover agents perform best-first search (BFS) following the `BFS-Prover-V1` implementation (Xin et al., 2025). Unless stated otherwise, we use a sampling temperature of 1.3, an expansion width of 3, and a length-normalization factor of 2.0 during expert iterations. For the planner, we use Gemini 2.5 Pro; other general-purpose reasoning models can reach similar performance given suitable prompts. Planner prompts are provided in Section C.2.

### 3.2 BENCHMARK RESULTS

We evaluated `BFS-Prover-V2` on two primary benchmarks: miniF2F Zheng et al. (2021), which targets high-school mathematical Olympiad problems (in-distribution), and ProofNet Azerbayev

et al. (2023), which covers undergraduate textbook level mathematics and serves as a rigorous test for out-of-distribution (OOD) generalization.

As detailed in Table 1, our system establishes a new state of the art among step-level tree-search provers. On the miniF2F-test set, our 32B model with the planner achieves an accuracy of 95.08% (95.49% on miniF2F-valid), while the 7B model reaches 92.6%.

Crucially, our approach demonstrates robust OOD generalization on ProofNet-test. The 32B model achieves 41.4%, and the 7B model reaches 34.4% via pure tree search, notably surpassing the 671B DeepSeek-Prover-V2 (37.1%) despite being 20 times smaller, and its 7B variant (29.6%), respectively. We attribute this superior OOD performance to the inherent flexibility of step-level proving: unlike whole-proof generation models that rely heavily on the training distribution, a trained step-prover can adapt its exploration strategy by adjusting search parameters at test time to match problem distributions, enabling effective transfer without retraining.

| Prover Method | budget | miniF2F-test | miniF2F-valid | ProofNet-test |
|---|---|---|---|---|
| *Tree-search provers* | | | | |
| InternLM2.5-StepProver-7B Wu et al. (2024a) | $256 \times 32 \times 600$ | 65.9% | 69.6% | $\approx 27\%$ |
| Hunyuan-Prover-7B Li et al. (2024b) | $600 \times 8 \times 400$ | 68.4% | - | - |
| BFS-Prover-V1-7B Xin et al. (2025) | $2048 \times 2 \times 600$ | 70.8% | - | - |
| | accumulative | 73.0% | - | - |
| MPS-Prover-7B[†] Liang et al. (2025) | $64 \times 4 \times 800 \times 8$ | 72.54% | - | - |
| | accumulative | 75.8% | - | - |
| BFS-Prover-V2-7B (this work) | accumulative | 82.4% | - | 34.4% |
| w/ Planner | accumulative | 92.6% | - | - |
| BFS-Prover-V2-32B (this work) | accumulative | 86.1% | 85.5% | 41.4% |
| w/ Planner | accumulative | 95.1% | 95.5% | - |
| *Whole-proof provers* | | | | |
| DeepSeek-Prover-V2-7B Ren et al. (2025) | 8192 / - / 1024 | 82.0% | - | 29.6% |
| w/ Prover Agent Baba et al. (2025) | 260 | 82.8% | - | - |
| DeepSeek-Prover-V2-671B Ren et al. (2025) | 8192 / - / 1024 | 88.9% | 90.6% | 37.1% |
| Kimina-Prover-72B[†] Wang et al. (2025) | 1024 | 87.7% | - | - |
| w/ TTRL search | accumulative | 92.2% | - | - |
| Goedel-Prover-V2-7B[†] Lin et al. (2025b) | 8192 | 90.2% | - | - |
| w/ Prover Agent Baba et al. (2025) | 260 | 86.5% | - | - |
| Goedel-Prover-V2-32B[†] Lin et al. (2025b) | 8192 | 92.2% | - | - |
| w/ Self-correction | 1024 | 92.6% | - | - |
| Delta-Prover[†] Zhou et al. (2025) | accumulative | 95.9% | - | - |
| Seed-Prover[†] Chen et al. (2025) | accumulative | 99.6% | - | - |

Table 1: Comparison with other leading theorem provers. [†] denotes concurrent work.

### 3.3 FURTHER ANALYSIS ON TRAINING

We report training-time ablations that justify the utility of tactic-level data curation, the critical role of periodic retraining in escaping local optima, and the motivation for scaling to larger base models.

**Perplexity-based tactic filtering:** We investigated the impact of our filtering strategy during the multi-stage expert iterations. We conducted an ablation on Checkpoint 3 in Figure 4 (derived after the first retraining phase). The training corpus consisted of 459,540 pairs of human data. Without tactic filtering, the combination of human data and synthetic expert iteration data yielded 857,897 state-action pairs. Training on this unfiltered set resulted in a validation loss of 0.5597 and a miniF2F test score of 75%. In contrast, applying our perplexity-based filtering reduced the dataset to 660,254 high-quality samples. Despite the smaller data volume, this filtered run (Checkpoint 4) resulted in a higher validation loss (0.6211) but a superior test score of 76.63%. Further validation on checkpoint 4 confirmed the approach's effectiveness: training with all expert iteration data resulted in performance degradation to 75.81%, while continued filtering improved performance to 77.04%.

**Periodic retraining:** Performance plateaus appeared at several checkpoints during training, where continued expert iteration did not improve and sometimes even reduced performance. Our periodic retraining mechanism consistently overcame these local optima. At checkpoint 2, performance stagnated at 75.41 percent. After retraining, accuracy briefly decreased to 75.00% but then increased to 76.64% in the next iteration. A similar pattern occurred at checkpoint 6: accuracy was 78.28% before retraining, declined to 77.05% immediately after retraining, and then reached 79.92% after two additional iterations. Crucially, this phenomenon is scalable: the larger 32B model exhibited a similar plateau at Checkpoint 19 (85.25%), dropped to 84.02% after retraining, and later reached 86.07%. Figure 4 presents the corresponding progression over time.

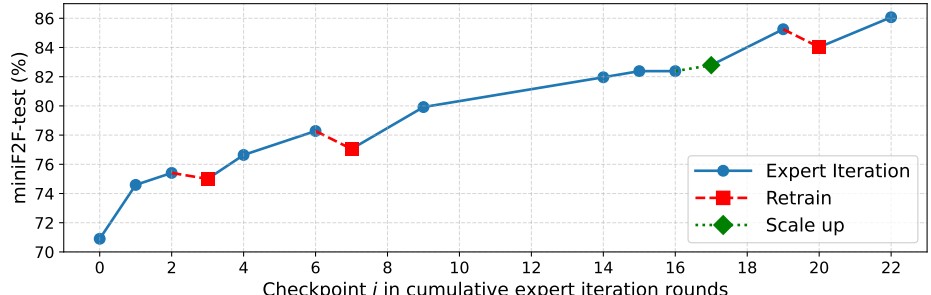

Figure 4: Sustained Performance Improvement through Expert Iteration and Periodic Retraining. This graph plots the prover's performance on the miniF2F benchmark vs. the expert iteration rounds.

**Model scaling:** We observed diminishing returns for the 7B prover as its improvement rate on the training corpus and miniF2F began to saturate, suggesting a capacity limitation inherent to the model size. To verify the scalability of our training recipe, we extended the multi-stage expert iteration experiments to the Qwen2.5-32B model. Scaling to 32B parameters yielded immediate gains even without additional data: checkpoints 16 and 17, trained on identical corpus, achieved 82.38% and 82.79% respectively. Critically, the 32B model demonstrated more superior out-of-distribution generalization on ProofNet (41.4% vs. 34.4% for 7B), indicating that increased model capacity enhances transfer to novel mathematical domains beyond the training distribution.

### 3.4 FURTHER ANALYSIS ON INFERENCE

**Computational budget:** We report the accumulative performance across a small grid of search hyperparameters (varying branching factors and depth rewards) with a budget of pass@8192 per configuration. We adopt this metric to rigorously estimate the system's maximum reasoning capability. Notably, our system remains robust without this accumulation: a single fixed configuration (branching factor 3 with depth reward 2 for miniF2F, or branching factor 8 with depth reward 1 for ProofNet) yields comparable results given a budget of pass@16384. The accumulative metric thus serves to capture the long tail of complex problems that reside at the boundaries of standard search parameters, providing a comprehensive view of the model's reachable proof set.

**Planner effectiveness:** The integration of the Planner agent provides a massive performance boost both model scales. The 7B model's performance jumps from 82.4% to 92.6% with the planner, while the 32B model improves from 86.1% to 95.1%. Notably, the planner narrows the performance

gap between model scales, demonstrating that effective search strategies can partially compensate for raw model capacity.

**Subgoal cache:** The shared subgoal cache is critical for reducing computational complexity. It effectively transforms the search complexity from the product of individual subgoal search spaces to their sum. Empirically, for `amc12a_2008_p25` and `mathd_algebra_17` in miniF2F-test, a planner-only setup without caching failed to find a solution after 8,192 cumulative prover instances. In contrast, with the shared subgoal cache, the system consistently solved these problems using fewer than 512 cumulative instances, preventing redundant computation on established subgoals.

## 4 Conclusion

The primary contributions of this work are the design, implementation, and empirical validation of a holistic system for scaling LLM-based step-provers. On the training side, our multi-stage expert iteration pipeline overcomes common performance plateaus and enables sustained improvement over an extended training period. On the inference side, by leveraging a planner agent for subgoal decomposition and a shared subgoal cache for collaborative search, our system transforms intractable search spaces into manageable sequences of tasks. Empirically, `BFS-Prover-V2` not only achieves state-of-the-art performance on miniF2F but also exhibits robust OOD generalization on ProofNet, providing strong evidence for the efficacy of our approach.

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

# A CASE STUDIES

## A.1 PROOF CONCISENESS AND TACTIC PROFICIENCY

A primary advantage of our step-level proof approach over the whole-proof paradigm is a dramatic reduction in proof length, which arises from the interactive nature of our method. By engaging with the Lean environment step by step, our model captures and leverages fine-grained tactic state information. This iterative feedback loop significantly improves its ability to employ powerful, high-level tactics such as simp, linarith, and nlinarith, enabling it to solve complex goals in a single step rather than through lengthy, explicit derivations.

To illustrate this contrast, we present a comparative analysis of proofs for two problems, aime_1984_p7 and amc12_2000_p1, generated by BFS-Prover-V2 and DeepSeek-Prover-V2.

---

**aime_1984_p7**

For the problem aime_1984_p7, BFS-Prover-V2 produces a remarkably concise proof, where a single tactic simp [$h_1$, $h_0$] effectively replaces over 2000 lines that appear in numerous have blocks within DeepSeek-Prover-V2's proof.

**BFS-Prover-V2 (step-level proof):**

```
theorem aime_1984_p7
  (f : ℤ → ℤ)
  (h₀ : ∀ n, 1000 ≤ n → f n = n - 3)
  (h₁ : ∀ n, n < 1000 → f n = f (f (n + 5))) :
  f 84 = 997 := by
  simp [h₁, h₀]
```

**DeepSeek-Prover-V2 (whole-proof):**

```
theorem aime_1984_p7 (f : ℤ → ℤ) (h₀ : ∀ n, 1000 ≤ n → f n = n - 3)
    (h₁ : ∀ n, n < 1000 → f n = f (f (n + 5))) : f 84 = 997 := by
  have h₂ : f 1004 = 1001 := by
    have h₂₁ : f 1004 = 1004 - 3 := by
      apply h₀
      <;> norm_num
    rw [h₂₁]
    <;> norm_num

  have h₃ : f 1003 = 1000 := by
    have h₃₁ : f 1003 = 1003 - 3 := by
      apply h₀
      <;> norm_num
    rw [h₃₁]
    <;> norm_num

  have h₄ : f 1002 = 999 := by
    have h₄₁ : f 1002 = 1002 - 3 := by
      apply h₀
      <;> norm_num
    rw [h₄₁]
    <;> norm_num

  -- ... (approximately 2000 lines of similar calculations omitted)
    ...
```

```
    have h₁₉₃ : f 89 = 998 := by
      have h₁₉₃₁ : f 89 = f (f (89 + 5)) := by
        apply h₁
        <;> norm_num
      rw [h₁₉₃₁]
      have h₁₉₃₂ : f (89 + 5) = f 94 := by norm_num
      rw [h₁₉₃₂]
      rw [h₁₉₂]
      <;> norm_num
      <;> linarith

    have h₁₉₄ : f 84 = 997 := by
      have h₁₉₄₁ : f 84 = f (f (84 + 5)) := by
        apply h₁
        <;> norm_num
      rw [h₁₉₄₁]
      have h₁₉₄₂ : f (84 + 5) = f 89 := by norm_num
      rw [h₁₉₄₂]
      rw [h₁₉₃]
      <;> norm_num
      <;> linarith

    rw [h₁₉₄]
    <;> norm_num
```

### amc12_2000_p1

For the problem `amc12_2000_p1`, `BFS-Prover-V2` finishes the proof with a concise application of `nlinarith`, replacing the roughly 80 lines of explicit case analysis over the unknowns *i*, *m*, and *o* required in `DeepSeek-Prover-V2`'s proof.

**BFS-Prover-V2 (step-level proof):**

```
theorem amc12_2000_p1
  (i m o : ℕ)
  (h₀ : i ≠ m ∧ m ≠ o ∧ o ≠ i)
  (h₁ : i*m*o = 2001) :
  i+m+o ≤ 671 := by
  obtain ⟨hi, hm, h₂⟩ := h₀
  obtain _ | _ | _ | i := i <;> obtain _ | _ | _ | m := m <;> obtain
  _ | _ | _ | o := o <;>try omega
  any_goals
    ring_nf at h₁ ⊢; nlinarith;
  ring_nf at * <;> try omega
  nlinarith [mul_nonneg (by positivity : 0 ≤ i) (by positivity : 0 ≤
    o), mul_nonneg (by positivity : 0 ≤ m) (by positivity : 0 ≤ i),
    mul_nonneg (by positivity : 0 ≤ o) (by positivity : 0 ≤ m)]
```

**DeepSeek-Prover-V2 (whole-proof):**

```
theorem amc12_2000_p1 (i m o : ℕ) (h₀ : i ≠ m ∧ m ≠ o ∧ o ≠ i) (h₁ :
    i * m * o = 2001) :
    i + m + o ≤ 671 := by

    -- ... (approximately 80 lines of properties related to unknowns
    i, m, and o omitted) ...
```

```
have h_main : i + m + o ≤ 671 := by
  have h₈ : i = 1 ∨ i = 3 ∨ i = 23 ∨ i = 29 ∨ i = 69 ∨ i = 87 ∨ i =
   667 ∨ i = 2001 := by
    have h₈₁ : i | 2001 := h₂
    have h₈₂ : i ≤ 2001 := h₅
    have h₈₃ : i > 0 := h_pos_i
    have h₈₄ : i = 1 ∨ i = 3 ∨ i = 23 ∨ i = 29 ∨ i = 69 ∨ i = 87 ∨
   i = 667 ∨ i = 2001 := by
      have h₈₅ : i | 2001 := h₈₁
      have h₈₆ : i ≤ 2001 := h₈₂
      interval_cases i <;> norm_num at h₈₅ ⊢ <;> omega
    exact h₈₄
  have h₉ : m = 1 ∨ m = 3 ∨ m = 23 ∨ m = 29 ∨ m = 69 ∨ m = 87 ∨ m =
   667 ∨ m = 2001 := by
    have h₉₁ : m | 2001 := h₃
    have h₉₂ : m ≤ 2001 := h₆
    have h₉₃ : m > 0 := h_pos_m
    have h₉₄ : m = 1 ∨ m = 3 ∨ m = 23 ∨ m = 29 ∨ m = 69 ∨ m = 87 ∨
   m = 667 ∨ m = 2001 := by
      have h₉₅ : m | 2001 := h₉₁
      have h₉₆ : m ≤ 2001 := h₉₂
      interval_cases m <;> norm_num at h₉₅ ⊢ <;> omega
    exact h₉₄
  have h₁₀ : o = 1 ∨ o = 3 ∨ o = 23 ∨ o = 29 ∨ o = 69 ∨ o = 87 ∨ o
   = 667 ∨ o = 2001 := by
    have h₁₀₁ : o | 2001 := h₄
    have h₁₀₂ : o ≤ 2001 := h₇
    have h₁₀₃ : o > 0 := h_pos_o
    have h₁₀₄ : o = 1 ∨ o = 3 ∨ o = 23 ∨ o = 29 ∨ o = 69 ∨ o = 87 ∨
    o = 667 ∨ o = 2001 := by
      have h₁₀₅ : o | 2001 := h₁₀₁
      have h₁₀₆ : o ≤ 2001 := h₁₀₂
      interval_cases o <;> norm_num at h₁₀₅ ⊢ <;> omega
    exact h₁₀₄
  rcases h₈ with (rfl | rfl | rfl | rfl | rfl | rfl | rfl | rfl)
  <;>
  rcases h₉ with (rfl | rfl | rfl | rfl | rfl | rfl | rfl | rfl)
  <;>
  rcases h₁₀ with (rfl | rfl | rfl | rfl | rfl | rfl | rfl | rfl)
  <;>
  norm_num [mul_assoc, mul_comm, mul_left_comm] at h₁ h₀ ⊢ <;>
  (try omega) <;>
  (try
    {
      norm_num at h₀ ⊢ <;>
      (try omega) <;>
      (try
        {
          ring_nf at h₁ ⊢ <;>
          omega
        })
    }) <;>
  (try
    {
      norm_num at h₀ ⊢ <;>
      (try omega) <;>
      (try
        {
          ring_nf at h₁ ⊢ <;>
          omega
        })
    }) <;>
```

```
      (try
        {
          norm_num at h₀ ⊢ <;>
          (try omega) <;>
          (try
            {
              ring_nf at h₁ ⊢ <;>
              omega
            })
        }) <;>
      (try
        {
          norm_num at h₀ ⊢ <;>
          (try omega) <;>
          (try
            {
              ring_nf at h₁ ⊢ <;>
              omega
            })
        })
      <;>
      (try omega)
      <;>
      (try
        {
          ring_nf at h₁ ⊢ <;>
          omega
        })

  exact h_main
```

A.2  NOVEL PROOF STRATEGIES

Another significant advantage of our step-level proof approach is its ability to discover novel proof strategies that whole-proof or human-proof methods typically would not consider. By exploring the proof space progressively, our system can identify non-obvious connections and construct solutions that are both elegant and insightful.

We illustrate this capability by examining the problems `imo_1963_p5` and `algebra_amgm_sum1toneqn_prod1tonleq1`, each of which highlights a distinct advantage of our approach.

---

**imo_1963_p5**

For the problem `imo_1963_p5`, our model, DeepSeek-Prover-V2, and Compfiles dataset provide step-level proof, whole-proof, and human-proof versions, respectively. Notably, both whole-proof and human-proof approaches employ similar strategies: multiplying both sides of the equation by $2 \cdot \sin(\pi/7)$, then applying sum-to-product trigonometric identities for simplification. In contrast, `BFS-Prover-V2` develops an entirely different approach: first transforming the left side of the equation into a polynomial in $\cos(\pi/7)$ using double and triple angle formulas, then proving that $\cos(\pi/7)$ satisfies the corresponding polynomial equation.

**BFS-Prover-V2 (step-level proof):**

```
theorem imo_1963_p5 :
  Real.cos (π / 7) - Real.cos (2 * π / 7) + Real.cos (3 * π / 7) = 1
    / 2 := by
  have x : Real.pi / 7 = Real.pi / 7 * 1 := by ring
```

```
864    have h : 3 * Real.pi / 7 = Real.pi - 4 * Real.pi / 7 := by ring
865    rw [h, cos_sub] <;> norm_num
866    have h2 := cos_two_mul (Real.pi / 7)
867    have h3 := cos_three_mul (π / 7)
868    rw [show 4 * Real.pi / 7 = Real.pi - 3 * Real.pi / 7 by ring,
         cos_sub]
869    simp [h2, h3, cos_two_mul, sin_pi, cos_pi]
870    ring_nf at h2 h3 ⊢
871    norm_num [h2, h3, cos_pi_div_two]
872    ring_nf
873      <;> have h4 := cos_pi
         <;> simp [h4]
874    ring_nf at * <;> norm_num
875    rw [← sub_eq_zero]
876    nth_rewrite 1 [← sub_eq_zero]
877    ring_nf
878    apply eq_of_sub_eq_zero
879    let y := cos (Real.pi * (1 / 7))
     have:= cos_three_mul (Real.pi * (1 / 7))
880    ring_nf at *
881    apply eq_of_sub_eq_zero
882    clear this h3 h2
883    apply eq_of_sub_eq_zero
     have := cos_three_mul (Real.pi * (1 / 7))
884    field_simp [mul_assoc] at *
885    on_goal 1 => ring
886    replace : Real.pi * (1 / 7 : ℝ) = Real.pi / 7 := by ring
887    try rw [this]; norm_num
888    have h5 := cos_three_mul (Real.pi / 7)
     have : 3 * (Real.pi / 7) = Real.pi - 4 * (Real.pi / 7) := by ring
889    simp [this, cos_pi] at h5
890    let z := cos (Real.pi / 7)
891    rcases lt_trichotomy 0 z with hz | hz | hz
892    any_goals simp_all [show cos (4 * (Real.pi / 7)) = cos (2 * (2 * (
       Real.pi / 7))) by ring,
893     cos_two_mul]
894    any_goals nlinarith [cos_sq_add_sin_sq (Real.pi / 7), Real.
       sin_pi_div_two_sub, pow_two_nonneg (cos (Real.pi / 7) - 1),
895     pow_two_nonneg (cos (Real.pi / 7) + 1)]
896    contrapose hz
897    refine not_lt.2 ?_
898    apply cos_nonneg_of_mem_Icc <;> constructor <;> linarith [pi_pos]
899
900
901
902    DeepSeek-Prover-V2 (whole-proof):
903
904    theorem imo_1963_p5 : Real.cos (Real.pi / 7) - Real.cos (2 * Real.pi
905      / 7) + Real.cos (3 * Real.pi / 7) = 1 / 2 := by
906      have h1 : 2 * Real.sin (Real.pi / 7) * (Real.cos (Real.pi / 7) -
907        Real.cos (2 * Real.pi / 7) + Real.cos (3 * Real.pi / 7)) = Real.
908        sin (Real.pi / 7) := by
909        have h1_1 : 2 * Real.sin (Real.pi / 7) * Real.cos (Real.pi / 7) =
910        Real.sin (2 * Real.pi / 7) := by
911          have h1_11 : Real.sin (2 * Real.pi / 7) = 2 * Real.sin (Real.pi
912      / 7) * Real.cos (Real.pi / 7) := by
913            have h1_12 : Real.sin (2 * Real.pi / 7) = Real.sin (2 * (Real.
914        pi / 7)) := by ring
915            rw [h1_12]
916            have h1_13 : Real.sin (2 * (Real.pi / 7)) = 2 * Real.sin (Real
917        .pi / 7) * Real.cos (Real.pi / 7) := by
             rw [Real.sin_two_mul]
             <;> ring
```

```
        rw [h1₁₃]
        <;> ring
      linarith
    have h1₂ : 2 * Real.sin (Real.pi / 7) * Real.cos (2 * Real.pi /
    7) = Real.sin (3 * Real.pi / 7) - Real.sin (Real.pi / 7) := by

 -- ... (approximately 20 lines of calculations omitted) ...

    have h1₃ : 2 * Real.sin (Real.pi / 7) * Real.cos (3 * Real.pi /
    7) = Real.sin (4 * Real.pi / 7) - Real.sin (2 * Real.pi / 7) :=
    by

 -- ... (approximately 20 lines of similar calculations omitted) ...

    have h1₄ : Real.sin (4 * Real.pi / 7) = Real.sin (3 * Real.pi /
    7) := by

 -- ... (approximately 20 lines of similar calculations omitted) ...

  have h2 : Real.sin (Real.pi / 7) > 0 := by
    apply Real.sin_pos_of_pos_of_lt_pi
    · linarith [Real.pi_pos, Real.pi_gt_three]
    · linarith [Real.pi_pos, Real.pi_gt_three]

  have h3 : Real.cos (Real.pi / 7) - Real.cos (2 * Real.pi / 7) +
    Real.cos (3 * Real.pi / 7) = 1 / 2 := by
    have h3₁ : 2 * Real.sin (Real.pi / 7) > 0 := by linarith
    have h3₂ : Real.cos (Real.pi / 7) - Real.cos (2 * Real.pi / 7) +
    Real.cos (3 * Real.pi / 7) = 1 / 2 := by
      apply mul_left_cancel₀ (show (2 * Real.sin (Real.pi / 7) : ℝ) ≠
     0 by linarith)
      nlinarith [Real.sin_le_one (Real.pi / 7), Real.sin_le_one (2 *
    Real.pi / 7), Real.sin_le_one (3 * Real.pi / 7),
        Real.sin_le_one (4 * Real.pi / 7), Real.sin_le_one (Real.pi /
     7)]
    exact h3₂

  apply h3
```

**Compfiles dataset (human-proof):**

```
theorem imo1963_p5 :
    Real.cos (π/7) - Real.cos (2*π/7) + Real.cos (3*π/7) = 1/2 := by
  rw [show (2*π/7) = π - (5*π/7) by linarith]
  rw [Real.cos_pi_sub]
  simp only [sub_neg_eq_add]
  have h : 2 * Real.sin (π / 7) ≠ 0 := by
    simp only [ne_eq, mul_eq_zero, OfNat.ofNat_ne_zero, false_or]
    apply ne_of_gt
    apply Real.sin_pos_of_pos_of_lt_pi
    simp only [Nat.ofNat_pos, div_pos_iff_of_pos_right, Real.pi_pos]
    trans 1
    · rw [div_lt_one (by linarith only)]
      linarith only [Real.pi_le_four]
    · linarith only [Real.pi_gt_three]
  apply (mul_right_inj' h).mp
  rw [left_distrib, left_distrib]
  have prod_sum : ∀ (x y : ℝ),
      2 * Real.sin x * Real.cos y = Real.sin (x + y) - Real.sin (y -
    x) := by
    intro x y
```

```
   rw [Real.sin_add, Real.sin_sub]
    linarith only
  rw [prod_sum, prod_sum, prod_sum]
  rw [show (π / 7 + π / 7)     = 2 * π / 7 by linarith only]
  rw [show (π / 7 - π / 7)     = 0        by linarith only]
  rw [show (π / 7 + 5 * π / 7) = 6 * π / 7 by linarith only]
  rw [show (5 * π / 7 - π / 7) = 4 * π / 7 by linarith only]
  rw [show (π / 7 + 3 * π / 7) = 4 * π / 7 by linarith only]
  rw [show (3 * π / 7 - π / 7) = 2 * π / 7 by linarith only]
  rw [Real.sin_zero]
  ring_nf
  rw [← Real.sin_pi_sub]
  rw [show (π - π * (6 / 7)) = π / 7 by linarith]
  congr
  linarith
```

## algebra_amgm_sum1toneqn_prod1tonleq1

For the problem algebra_amgm_sum1toneqn_prod1tonleq1, the whole-proof model
DeepSeek-Prover-V2 adopts a standard, first-principles approach: it proceeds by man-
ually handling cases ($n = 0$, some $a_i = 0$, all $a_i > 0$), and then takes the logarithm of the
product and then applies the well-known inequality $\ln(x) \leq x - 1$ to each term, resulting in a
verbose proof. In contrast, BFS-Prover-V2 recognizes the problem as a special case of the
Arithmetic Mean-Geometric Mean (AM-GM) inequality. It directly invokes the corresponding
theorem from Mathlib, Real.geom_mean_le_arith_mean, demonstrating an ability to
leverage high-level library theorems for a more insightful and efficient proof.

### BFS-Prover-V2 (step-level proof):

```
theorem algebra_amgm_sum1toneqn_prod1tonleq1
  (a : ℕ → NNReal)
  (n : ℕ)
  (h₀ : ∑ x in Finset.range n, a x = n) :
  ∏ x in Finset.range n, a x ≤ 1 := by
  have g := h₀
  revert h₀
  intro amgm
  let S := Finset.range n
  by_cases h1 : n = 0
  simp[h1]
  have hn  : 0 < n := by omega
  let μ := (fun (x : ℕ) => (a x : ℝ))
  let w : ℕ → ℝ := fun _ => 1
  have w_nonneg : ∀ i, 0 ≤ w i
  simp [w, zero_le_one]
  have w_pos  : 0 < ∏ i in Finset.range n, w i
  all_goals aesop (add simp w)
  have w_nonneg : ∀ x ∈ S, 0 ≤ w x
  any_goals simp [w]
  have w_pos  : 0 < ∏ x ∈ S, w x
  simp [w]
  have amgm_real  : (∏ x ∈ S, μ x ^ (w x : ℝ)) ^ ((∑ x ∈ S, (w x : ℝ)
    )⁻¹) ≤ (∑ x ∈ S, (w x * μ x : ℝ)) / (∑ x ∈ S, (w x : ℝ))
  apply Real.geom_mean_le_arith_mean
  exact w_nonneg
  simp [w, S, hn]
  field_simp[μ]
  simp [μ, w, S] at amgm_real
```

```
norm_cast at amgm_real
simp[ amgm] at amgm_real
by_cases H : ∏ a_1 ∈ Finset.range n, a a_1 = 0
simp [H]
simp [← NNReal.coe_le_coe, amgm] at amgm_real
rw  [div_self] at amgm_real
apply le_of_not_lt
all_goals aesop (add simp NNReal)
rw[←NNReal.coe_one] at amgm_real
norm_cast at amgm_real
contrapose! amgm_real
erw [ Real.one_lt_rpow_iff]
left
all_goals aesop (add simp [Finset.range])
```

**DeepSeek-Prover-V2 (whole-proof):**

```
theorem algebra_amgm_sum1toneqn_prod1tonleq1 (a : ℕ → NNReal) (n : ℕ)
    (h₀ : ∑ x in Finset.range n, a x) = n) : (∏ x in Finset.range n,
    a x) ≤ 1 := by
  have h₁ : (∏ x in Finset.range n, (a x : ℝ)) ≤ 1 := by
    have h₂ : (∑ x in Finset.range n, (a x : ℝ)) = n := by
      exact_mod_cast h₀
    by_cases h₃ : n = 0
      simp [h₃]
      have h₄ : 0 < n := Nat.pos_of_ne_zero h₃
      by_cases h₅ : ∃ i, i ∈ Finset.range n ∧ (a i : ℝ) = 0
        obtain ⟨i, hi, h₆⟩ := h₅
        have h₇ : (∏ x in Finset.range n, (a x : ℝ)) = 0 := by
          have h₈ : i ∈ Finset.range n := hi
          have h₉ : (a i : ℝ) = 0 := h₆
          have h₁₀ : (∏ x in Finset.range n, (a x : ℝ)) = 0 := by
            calc
              (∏ x in Finset.range n, (a x : ℝ)) = ∏ x in Finset.
range n, (a x : ℝ) := rfl
              _ = 0 := by
                apply Finset.prod_eq_zero h₈
                simp [h₉]
          exact h₁₀
        linarith
        have h₅' : ∀ i, i ∈ Finset.range n → (a i : ℝ) > 0 := by
          intro i hi
          have h₆ : ¬∃ i, i ∈ Finset.range n ∧ (a i : ℝ) = 0 := h₅
          have h₇ : (a i : ℝ) ≠ 0 := by
            intro h
            have h₈ : ∃ i, i ∈ Finset.range n ∧ (a i : ℝ) = 0 := ⟨i,
hi, by simp_all⟩
            contradiction
          have h₈ : (a i : ℝ) > 0 := by
            have h₉ : (a i : ℝ) ≥ 0 := by exact_mod_cast (a i).prop
            contrapose! h₇
            linarith
          exact h₈
        have h₆ : (∏ x in Finset.range n, (a x : ℝ)) ≤ 1 := by
          have h₇ : ∑ x in Finset.range n, Real.log ((a x : ℝ)) ≤ 0
 := by
            have h₈ : ∑ x in Finset.range n, Real.log ((a x : ℝ)) ≤ ∑
 x in Finset.range n, ((a x : ℝ) - 1) := by
              apply Finset.sum_le_sum
              intro i hi
              have h₉ : (a i : ℝ) > 0 := h₅' i hi
```

```
              have h₁₀ : Real.log ((a i : ℝ)) ≤ (a i : ℝ) - 1 := by
                have h₁₁ : Real.log ((a i : ℝ)) ≤ (a i : ℝ) - 1 := by
                  linarith [Real.log_le_sub_one_of_pos h₉]
                exact h₁₁
              exact h₁₀

  -- ... (approximately 40 lines of calculations omitted) ...

  have h₂ : (∏ x in Finset.range n, a x) ≤ 1 := by
    have h₃ : (∏ x in Finset.range n, a x : ℝ) ≤ 1 := by
      exact h₁
    have h₄ : (∏ x in Finset.range n, a x : ℝ) = (∏ x in Finset.range
    n, a x : ℝ) := rfl
    have h₅ : (∏ x in Finset.range n, a x : ℝ) = (∏ x in Finset.range
    n, (a x : ℝ)) := by simp
    have h₆ : (∏ x in Finset.range n, a x : ℝ) ≤ 1 := by simpa [h₅]
    using h₁
    have h₇ : (∏ x in Finset.range n, a x : NNReal) ≤ 1 := by
      norm_cast at h₆ ⊢
      <;> simp_all [Finset.prod_range_succ]
      <;> norm_num
      <;> linarith
    simpa using h₇
  exact h₂
```

Despite the advantages of step-level proof in significantly reducing proof length and discovering novel proof strategies, step-level proof has one notable limitation: poor readability. The interactive nature of step-level proof generation often results in proofs that are more challenging for humans to follow and understand compared to the more structured and explanatory whole-proof approaches. This trade-off between conciseness and readability represents a crucial consideration when evaluating the practical utility of different proof generation paradigms.

## B  ILLUSTRATION OF PLANNER-PROVER PARADIGM WITH AN IMO PROBLEM

To demonstrate the effectiveness of our Planner-Prover paradigm, we present an analysis of the solution process for a challenging IMO problem: `imo_1969_p2`.

In the following proof, the statements `h_coeffs_polar`, `h_y_rewritten_with_polar`, and `h_y_collapsed_to_single_cos` represent the dynamic replanning phase, while all other `have` statements belong to the initial planning phase. Unlike in conventional whole-proof methods, `have` statements in our framework are presented without the `:= by` clause. This example highlights the crucial role of dynamic replanning in our system. Without dynamic replanning, the prover gets stuck at `h_y_is_sinusoid`, failing to complete the proof even after 7,200 attempts. With dynamic replanning, however, the system successfully completes the proof in just 800 attempts. The dynamic replanning process breaks down complex steps into smaller, more manageable subgoals, which enables the prover to bypass critical bottlenecks more efficiently.

---

**imo_1969_p2 - Part 1**

```
theorem imo_1969_p2
  (m n : ℝ)
  (k : ℕ)
  (a : ℕ → ℝ)
  (y : ℝ → ℝ)
  (h₀ : 0 < k)
  (h₁ : ∀ x, y x = ∑ i in Finset.range k, ((Real.cos (a i + x)) / (2^
    i)))
  (h₂ : y m = 0)
  (h₃ : y n = 0) : ∃ t : ℤ, m - n = t * Real.pi := by
  have h_cos_add : ∀ i x, Real.cos (a i + x) = Real.cos (a i) * Real.
    cos x - Real.sin (a i) * Real.sin x
  simp [cos_add, add_right_inj]

  have h_y_sum_expanded : ∀ (x : ℝ), y x = ∑ i in (Finset.range k :
    Finset ℕ), (Real.cos (a i) * Real.cos x - Real.sin (a i) * Real.
    sin x) / ((2 : ℕ) ^ i : ℝ)
  simp [h₁, h_cos_add ]

  have h_y_sum_split : ∀ (x : ℝ), y x = ∑ i in (Finset.range k :
    Finset ℕ), Real.cos (a i) * Real.cos x / ((2 : ℕ) ^ i : ℝ)) - ∑
    i in (Finset.range k : Finset ℕ), Real.sin (a i) * Real.sin x /
    ((2 : ℕ) ^ i : ℝ))
  intro z <;> simp_rw [h_y_sum_expanded]
  simp [sub_div, Finset.sum_sub_distrib]

  have h_y_expand : ∀ (x : ℝ), y x = ∑ i in (Finset.range k : Finset
    ℕ), Real.cos (a i) / ((2 : ℕ) ^ i : ℝ)) * Real.cos x - ∑ i in (
    Finset.range k : Finset ℕ), Real.sin (a i) / ((2 : ℕ) ^ i : ℝ)) *
    Real.sin x
  intro x_exp
  simp only [Finset.sum_mul, h_y_sum_split]
  congr <;> symm <;> field_simp <;> ring

  have h_k_ge_one : 1 ≤ k
  apply Nat.succ_le_of_lt    <;> exact h₀

  have h_complex_repr : (∑ i in (Finset.range k : Finset ℕ), Real.
    cos (a i) / ((2 : ℕ) ^ i : ℝ), ∑ i in (Finset.range k : Finset ℕ)
    , Real.sin (a i) / ((2 : ℕ) ^ i : ℝ)) : ℂ) = ∑ i in (Finset.range
    k : Finset ℕ), Complex.exp (↑(a i) * Complex.I) / ↑(((2 : ℕ) ^ i
    ) : ℝ)
  simp [Complex.exp_mul_I, div_eq_inv_mul, Complex.ext_iff]
```

---

```
simp [Complex.cos_ofReal_re, Complex.sin_ofReal_re] <;> field_simp
  <;> norm_cast
constructor <;> apply Finset.sum_congr <;> aesop
field_simp [_root_.pow_add, show (4 : ℝ) = 2 ^ 2 by norm_num] <;>
  ring
norm_num [mul_comm _ 2, pow_mul]
rewrite [show (4 : ℝ) ^ x = (2 * 2 : ℝ) ^ x by ring, mul_pow]
  <;> field_simp
  <;> ring

have h_sum_split : ∑ i in (Finset.range k : Finset ℕ), Complex.exp
  (↑(a i) * Complex.I) / ↑(((2 : ℕ) ^ i) : ℝ)) = Complex.exp (↑(a
  0) * Complex.I) + ∑ i in (Finset.Icc 1 (k-1) : Finset ℕ), Complex
  .exp (↑(a i) * Complex.I) / ↑(((2 : ℕ) ^ i) : ℝ)
have h_range_split  : Finset.range k = insert 0 (Finset.Icc 1 (k -
  1))
ext x  <;>  simp  [Nat.lt_succ_iff]
rcases x with (_|_|x) <;> omega
rw [h_range_split, Finset.sum_insert]
norm_num [pow_zero, eq_self_iff_true]
simp [Nat.le_zero]

have h_abs_head : Complex.abs (Complex.exp (↑(a 0) * Complex.I)) =
  1
simp [Complex.abs_exp, eq_self_iff_true]

have h_tail_geom_sum_val : ∑ i in (Finset.Icc 1 (k - 1) : Finset ℕ
  ), 1 / ((2 : ℕ) ^ i : ℝ)) = 1 - 1 / (2 : ℝ) ^ (k - 1)
have h_tight  : (1 : ℝ) ≤ k
norm_cast at * <;>
linarith
clear h_tight h_sum_split h_complex_repr h_y_expand h_y_sum_split
  h_y_sum_expanded h_cos_add h₂ h₃ h₁ h₀
induction' k <;> simp [Finset.sum_Icc_succ_top, *]
induction' ⟨ℕ⟩ <;> simp_all [Finset.sum_Icc_succ_top, pow_succ]
ring
<;>ring_nf

have h_abs_tail_le : Complex.abs ∑ i in (Finset.Icc 1 (k-1) :
  Finset ℕ), Complex.exp (↑(a i) * Complex.I) / ↑(((2 : ℕ) ^ i) : ℝ
  )) ≤ 1 - 1 / (2 : ℝ) ^ (k - 1)
rw [← h_tail_geom_sum_val]
apply (Complex.abs.sum_le _ _).trans_eq
apply Finset.sum_congr rfl
intro i _
simp [Complex.abs_exp_ofReal_mul_I, Nat.cast_pow, Nat.cast_ofNat]

have h_abs_tail_lt_one : Complex.abs ∑ i in (Finset.Icc 1 (k-1) :
  Finset ℕ), Complex.exp (↑(a i) * Complex.I) / ↑(((2 : ℕ) ^ i) : ℝ
  )) < 1
refine lt_of_le_of_lt h_abs_tail_le ?_
refine sub_lt_self _ (by positivity)

have h_abs_ge_by_rev_triangle : Complex.abs ∑ i in (Finset.range k
  : Finset ℕ), Complex.exp (↑(a i) * Complex.I) / ↑(((2 : ℕ) ^ i)
  : ℝ)) ≥ 1 - Complex.abs ∑ i in (Finset.Icc 1 (k-1) : Finset ℕ),
  Complex.exp (↑(a i) * Complex.I) / ↑(((2 : ℕ) ^ i) : ℝ))
rw [h_sum_split]
rw [← h_abs_head]
apply Complex.abs.le_add
```

**imo_1969_p2 - Part 2**

```
have h_abs_ge_final : Complex.abs (∑ i in (Finset.range k : Finset
 ℕ), Complex.exp (↑(a i) * Complex.I) / ↑(((2 : ℕ) ^ i) : ℝ)) ≥ 1
 / (2 : ℝ) ^ (k-1)
refine' _root_.trans h_abs_ge_by_rev_triangle _
linarith [h_abs_tail_le]

have h_abs_gt_zero : 0 < Complex.abs (∑ i in (Finset.range k :
 Finset ℕ), Complex.exp (↑(a i) * Complex.I) / ↑(((2 : ℕ) ^ i) : ℝ
 ))
linarith [pow_two_nonneg ((k - 1 : ℕ) : ℝ) ]

have h_complex_val_ne_zero : (⟨∑ i in (Finset.range k : Finset ℕ),
 Real.cos (a i) / ((2 : ℕ) ^ i : ℝ), ∑ i in (Finset.range k :
 Finset ℕ), Real.sin (a i) / ((2 : ℕ) ^ i : ℝ)⟩ : ℂ) ≠ 0
focus all_goals (norm_num; aesop)
```

```
have h_coeffs_polar : ∃ (R b : ℝ), 0 < R ∧ (∑ i in (Finset.
 range k : Finset ℕ), Real.cos (a i) / ((2 : ℕ) ^ i : ℝ)) = R
 * Real.cos b ∧ (∑ i in (Finset.range k : Finset ℕ), Real.sin
 (a i) / ((2 : ℕ) ^ i : ℝ)) = R * Real.sin b
set x := ∑ i ∈ Finset.range k, cos (a i) / ((2 : ℝ) ^ i)
use Complex.abs (∑ i ∈ Finset.range k, Complex.exp (↑(a i) *
 Complex.I) / ↑(↑2 ^ i))
let y : ℝ := ∑ i ∈ Finset.range k, sin (a i) / 2^i
have h := Complex.abs_mul_cos_add_sin_mul_I (∑ i in Finset.
 range k, Complex.exp ((a i : ℝ) * Complex.I) / (2 : ℂ) ^ i)
use  Complex.arg (∑ i in Finset.range k, Complex.exp (↑(a i) *
 Complex.I) / (2:ℝ) ^ i)
simp_all [Complex.ext_iff]
```

```
have h_y_rewritten_with_polar : ∃ (R a : ℝ), 0 < R ∧ ∀ x, y x =
 R * Real.cos a * Real.cos x - R * Real.sin a * Real.sin x
obtain ⟨R, phi, hR_pos, h_cos_eq1, h_sin_eq1⟩ :=
 h_coeffs_polar
use R, phi <;> simp_all[Complex.exp_mul_I, Complex.abs]
```

```
have h_y_collapsed_to_single_cos : ∃ (R  a : ℝ), 0 < R ∧ ∀ x, y
 x = R * Real.cos (x + a)
rcases h_y_rewritten_with_polar with ⟨R, a', h_R_pos, h_y_⟩
use R, a', h_R_pos <;> intros <;> simp [h_y_, cos_add] <;> ring
```

```
have h_y_is_sinusoid : ∃ (R a : ℝ), 0 < R ∧ (∀ x, y x = R * Real.
 cos (x - a))
obtain ⟨ R, a, _, hy ⟩ := h_y_collapsed_to_single_cos
use R, -a <;> aesop

have h_roots_exist : ∃ (R a : ℝ), 0 < R ∧ y m = R * Real.cos (m - a
 ) ∧ y n = R * Real.cos (n - a)
```

```
rcases h_y_is_sinusoid with ⟨R, a, h_R_pos, h_y_R_a⟩
exact ⟨ R, a, h_R_pos,
  by simp [h_y_R_a], by simp [h_y_R_a] ⟩

have h_cos_zero : ∃ (R a : ℝ), 0 < R ∧ Real.cos (m - a) = 0 ∧ Real.
  cos (n - a) = 0
rcases h_roots_exist with ⟨R, a, h_rPos,  h_mEq,  h_nEq⟩
exact
  ⟨R, a, h_rPos,
    by have := h₂; have := h₃; field_simp [h₁] at * <;> nlinarith,
    by have := h₃; have := h₂; field_simp [h₁] at * <;> nlinarith⟩

have h_roots_in_pi_half_multiples : ∃ (a : ℝ) (t₁ t₂ : ℤ), m - a =
  (2 * (t₁ : ℝ) + 1) * Real.pi / 2 ∧ n - a = (2 * (t₂ : ℝ) + 1) *
  Real.pi / 2
rcases h_cos_zero with ⟨R, a, _, h_m_cos_zero, h_n_cos_zero⟩
rw [ cos_eq_zero_iff] at h_m_cos_zero  h_n_cos_zero
exact ⟨ a,  ↑( Classical.choose h_m_cos_zero ) , ↑( Classical.
  choose h_n_cos_zero ) , by convert h_m_cos_zero.choose_spec , by
  convert  h_n_cos_zero.choose_spec ⟩

have h_m_minus_n_form : ∃ t₁ t₂ : ℤ, m - n = ((2 * (t₁ : ℝ) + 1) *
  Real.pi / 2) - ((2 * (t₂ : ℝ) + 1) * Real.pi / 2)
obtain ⟨z, t₁, t₂, h_z_root_m, h_z_root_n⟩ :=
  h_roots_in_pi_half_multiples
refine ⟨t₁ , t₂,?_⟩<;>
linarith

have h_m_minus_n_simplified : ∃ t₁ t₂ : ℤ, m - n = (↑(t₁ - t₂) : ℝ)
   * Real.pi
rcases h_m_minus_n_form with ⟨t₁, t₂, h_form⟩  <;>
  exists t₁  <;> exists t₂  <;>  field_simp at h_form ⊢  <;>
  linarith

obtain ⟨t₁, t₂,h_m_sub_n_t₁_t₂⟩ := h_m_minus_n_simplified  <;>  use
   t₁ - t₂  <;>  linarith [h_m_sub_n_t₁_t₂]
```

# C  PROMPTS USED IN THIS WORK

## C.1  PROMPTS FOR AUTOFORMALIZATION

Our autoformalization pipeline operates in two stages to ensure syntactic correctness. First, an `Initial Formalization Prompt` (shown below) translates a natural language problem into a Lean 4 theorem statement. If the generated code fails to compile, an `Error Feedback Prompt` is then deployed to revise the statement, using the verbatim error message from the Lean compiler as direct feedback for revision.

---

**Prompt for Initial Formalization**

You are an expert in math proof and the theorem prover: Lean. Given a math problem that contains the question and all conditions, and its corresponding solution that contains solution steps and the correct answer, generate a mathematically equivalent proof problem and rewrite it in the Lean 4 statement. You should follow the following procedures.

   a): Identify all questions and conditions in the given problem.

   b): Identify all solution steps and the correct answers in the given solution.

   c): With the questions and conditions in a) and correct answers in b), translate the (question, conditions, correct answer) tuple to a mathematically equivalent proof problem that proves question == answer given conditions.

   d): Rewrite the math proof problem in c) to a Lean 4 statement. Note that you should write the statement only, no proof is required. This also means you do not need to consider the solution steps either.

The first priority is to ensure the generated Lean code can be built successfully. Consider using the following tips.

   • Use a broader import, e.g., `import Mathlib`, to bring in the entirety of the necessary library, and remove specific import of submodules, e.g., `import Mathlib.LinearAlgebra.BasicReal3`, accordingly.

   • Add `noncomputable` before `def` only when necessary.

   • Use `by` instead of `begin end`.

   • Add `sorry` to skip the proof.

You should strictly follow the below criteria to guarantee the lean statement is equivalent to the mathematical problem.

   • Each definition used in Lean 4 statement should only directly appear in the conditions problem in a).

   • Each definition should NOT come from and assume any knowledge directly from the solution step in b).

   • Each condition in a) should be used as a definition in Lean 4.

   • For any implications appearing in the conclusions of the original problem, extract their antecedents and declare them as explicit assumptions before the colon, leaving only the consequent in the conclusion after the colon.

   • For equations, structure the theorem in the form 'conditions : conclusions' where conditions include variable definitions and domains, and conclusions are the solutions to the equation, avoiding implication or equivalence symbols.

**Below are examples to illustrate the process:**

---

**Example 1 (Number Theory):**
**Lean 4 statement:**

```
theorem nt3_problem (n p : ℕ) (hn : n > 1) (hp : Nat.Prime p)
  (h1 : n | (p - 1)) (h2 : p | (n^6 - 1)) :
  ∃ k : ℕ, (p - n = k^2) ∨ (p + n = k^2) := by
  sorry
```

**problem:**
NT3. Let $n > 1$ be a positive integer and $p$ a prime number such that $n \mid (p - 1)$ and $p \mid (n^6 - 1)$. Prove that at least one of the numbers $p - n$ and $p + n$ is a perfect square.

**Example 2 (Number Theory):**
**Lean 4 statement:**

```
theorem nt4_problem (x y : ℕ)
  (hx : x > 0) (hy : y > 0)
  (h1 : ∃ m : ℕ, 3 * x + 4 * y = m^2)
  (h2 : ∃ n : ℕ, 4 * x + 3 * y = n^2) :
  7 | x ∧ 7 | y := by
  sorry
```

**problem:**
NT4. If the positive integers $x$ and $y$ are such that both $3x + 4y$ and $4x + 3y$ are perfect squares, prove that both $x$ and $y$ are multiples of 7.

**Example 3 (Algebra):**
**Lean 4 statement:**

```
theorem sum_not_zero (a b c d : ℝ)
  (h1 : a * b * c - d = 1)
  (h2 : b * c * d - a = 2)
  (h3 : c * d * a - b = 3)
  (h4 : d * a * b - c = -6) :
  a + b + c + d ≠ 0 := by
  sorry
```

**problem:**
The real numbers $a, b, c, d$ satisfy simultaneously the equations $abc - d = 1, bcd - a = 2, cda - b = 3, dab - c = -6$. Prove that $a + b + c + d \neq 0$.

**Example 4 (Inequality):**
**Lean 4 statement:**

```
theorem inequality_proof (a b c : ℝ)
  (ha : a > 0) (hb : b > 0) (hc : c > 0) :
  8 / ((a + b)^2 + 4*a*b*c) +
  8 / ((b + c)^2 + 4*a*b*c) +
  8 / ((c + a)^2 + 4*a*b*c) +
  a^2 + b^2 + c^2 ≥
  8 / (a + 3) + 8 / (b + 3) + 8 / (c + 3) := by
  sorry
```

**problem:**
The real numbers $a, b, c, d$ satisfy simultaneously the equations $abc - d = 1, bcd - a = 2, cda - b = 3, dab - c = -6$. Prove that $a + b + c + d \neq 0$.

Now, use the same process for the following problem and solution:

{**problem**}

{**solution**}

---

**Prompt for Error Feedback**

You are an expert in math proof and the theorem prover: Lean. You are given the following math problem that contains the question and all conditions, and its corresponding solution that contains solution steps and the correct answer.

{**problem**}

{**solution**}

A mathematically equivalent proof problem that proves question == answer given conditions is generated and rewritten in the Lean 4 statement, as shown below:

{**Lean 4 statement**}

However, this lean code got error with `lake build`, and here is the error message:

{**error message**}

Please modify the lean code to ensure it can be built successfully with `lake build`. Here is a few tips that might help:

- Use a broader import, e.g., `import Mathlib`, to bring in the entirety of the necessary library, and remove specific import of submodules, e.g., `import Mathlib.LinearAlgebra.BasicReal3`, accordingly.
- Add `noncomputable` before `def` only when necessary.
- Use `by` instead of `begin end`.
- Add `sorry` to skip the proof.

---

## C.2 PROMPTS FOR PLANNER

**Prompt for Initial Planning**

You are an expert assistant specializing in Math Olympiads and the Lean 4 theorem prover. Your primary goal is to generate **syntactically perfect, type-checkable** Lean 4 intermediate step code snippets (**plan**) for a given theorem. It is crucial to strictly adhere to the following rules—any violation will be considered an error.

**Task**
Given the following Lean 4 theorem tactic state, generate the core intermediate subgoals (`have` statements) needed for the proof.

**Mandatory Rules**
You must comply with every rule in this section. Failure to adhere to any single rule will result in an incorrect output.

1. **Critical Rule: Explicitly Specify Set/Finset Types**
   This is the most common and fatal point of error. You must explicitly declare the type for any `Set` or `Finset` literal. This rule is non-negotiable.

   ```
   - Correct: ({ {-1, 0, 1}} : Set ℤ)
   - Incorrect: { {-1, 0, 1}}
   ```

2. **Omit the Proof**: Never provide the proof. Only state the `have` statement itself.
3. **Valid Lean 4 Code**: The entire output block must be type-checkable in a Lean 4.10.0 environment.
4. **Use Existing Names**: Use the exact, existing lemma and definition names from `mathlib`. Do not invent names.

5. **No Undeclared Variables**: Do not introduce any variables or constants not declared in the original theorem statement.

6. **Explicit Multiplication**: Multiplication must always use the `*` symbol.

   ```
   – Correct: a * x
   – Incorrect: ax
   ```

7. **No Chained Inequalities**: Never use chained inequalities. They must be split using logical AND ∧.

   ```
   – Correct: a <= x ∧ x <= b
   – Incorrect: a <= x <= b
   ```

8. **Correct Logarithm Function**: `Real.log` is only for the natural logarithm. For logarithms with a specified base, you must use `Real.logb`.

   ```
   – Correct: Real.logb (2 : ℝ) 8
   – Incorrect: Real.log (2 : ℝ) 8
   ```

9. **Factorial Notation**: In Lean, factorials must be written as `(n)!` or `Nat.factorial n`, not `n!`.

   ```
   – Correct: (n)! or Nat.factorial n
   – Incorrect: n!
   ```

10. **Numeric Types Must Be Explicitly Annotated**: To avoid type ambiguity in Lean, any expression involving numeric operations must have at least one number's type specified.

    ```
    – For division: (1 : ℝ) / 2 = 0.5, but (1 : ℤ) / 2 = 0.
    – For subtraction: (1 : ℤ) – 2 = -1, but (1 : ℕ) – 2 = 0.
    – Correct: (a : ℝ) / b, a / (b : ℝ), (n : ℤ) – m
    – Incorrect: a / b, n – m
    ```

11. **Interval Notation**: Do not use `Icc`, `Ioo`, `Ico`, `Ioc`, etc., to represent intervals. Only use inequalities.

    ```
    – Correct: a <= x ∧ x <= b
    – Incorrect: Icc a b
    ```

12. **Complex Numbers**: Use `Complex.I` for the imaginary unit and `Complex.abs` for the modulus/absolute value of a complex number.

13. **Avoid Common Inequality Theorems**: Avoid using common inequality theorems like Holder's or Jensen's. For inequality problems, try to ensure each proof step only requires basic simplification.

14. **Proving Equivalences**: For proofs of equivalences (iff), ensure each `have` statement is an implication, where the antecedent is the left side of the equivalence (when proving left-to-right) or the right side (when proving right-to-left).

15. **Real.pi Notation**: Consistently use `Real.pi` instead of $\pi$.

16. **Final Check**: Before providing the plan, perform a final review to ensure you have scrupulously followed all the rules above, especially the critical rule regarding `Set`/`Finset`.

**Below are examples to illustrate the process:**

**Example 1:**
**Theorem:**

```
theorem singapore2019_r1_p7 (x : ℝ) (hx : Real.tan x = 5) :
  (6 + Real.sin (2 * x)) / (1 + Real.cos (2 * x)) = 83 := by
```

**Plan:**

```
have h₁ : Real.sin x = 5 * Real.cos x
have h₂ : Real.sin x ^ 2 = 25 * Real.cos x ^ 2
have h₃ : 26 * Real.cos x ^ 2 = 1
have hsin2x_val : Real.sin (2 * x) = (5 : ℝ) / (13 : ℝ)
have hcos2x_val : Real.cos (2 * x) = -(12 : ℝ) / (13 : ℝ)
```

**Example 2:**
**Theorem:**

```
theorem problem4
  (g : ℕ → ℝ)
  (h : ∀ k : ℕ, 5 ≤ k → k ≤ 124 → g k = (Real.logb (k : ℝ) ((7 : ℝ) ^
    (k ^ 2 - 1))) / (Real.logb ((k + 1 : ℝ)) ((7 : ℝ) ^ (k ^ 2 - 4))
    )) :
  (∏ k in Finset.Icc (5 : ℕ) 124, g k) = (41 : ℝ) / 7 := by
```

**Plan:**

```
have h_prod_split : (∏ k in (Finset.Icc 5 124 : Finset ℕ), g k) = (
  ∏ k in (Finset.Icc 5 124 : Finset ℕ), ((k ^ 2 - 1) / (k ^ 2 - 4 :
  ℝ))) * (∏ k in (Finset.Icc 5 124 : Finset ℕ), (Real.logb (k : ℝ)
  (7 : ℝ) / Real.logb ((k + 1 : ℝ)) (7 : ℝ)))
have h_telescope_part1 : (∏ k in (Finset.Icc 5 124 : Finset ℕ), ((k
  ^ 2 - 1) / (k ^ 2 - 4 : ℝ))) = (41 : ℝ) / 21
have h_telescope_part2 : (∏ k in (Finset.Icc 5 124 : Finset ℕ), (
  Real.logb (k : ℝ) (7 : ℝ) / Real.logb ((k + 1 : ℝ)) (7 : ℝ))) = 3
have h_final_product : (41 / 21 : ℝ) * 3 = (41 : ℝ) / 7
```

**Example 3:**
**Theorem:**

```
theorem amc12b_variant_p13
  (S : Finset ℝ)
  (h₀ : ∀ (x : ℝ), x ∈ S ↔ 0 < x ∧ x ≤ 2 * Real.pi ∧ 2 - 4 * Real.sin
    x + 3 * Real.cos (3 * x) = 0) :
  S.card = 4 := by
```

**Plan:**

```
have h_interval1 : ∃ x, 0 ≤ x ∧ x < Real.pi / 2 ∧ (2 - 4 * Real.sin
  x + 3 * Real.cos (3 * x) = 0)
have h_interval2 : ∃ x, Real.pi / 2 ≤ x ∧ x < 3 * Real.pi / 4 ∧ (2
  - 4 * Real.sin x + 3 * Real.cos (3 * x) = 0)
have h_interval3 : ∃ x, 3 * Real.pi / 4 ≤ x ∧ x < Real.pi ∧ (2 - 4
  * Real.sin x + 3 * Real.cos (3 * x) = 0)
have h_interval4 : ∃ x, Real.pi ≤ x ∧ x < 2 * Real.pi ∧ (2 - 4 *
  Real.sin x + 3 * Real.cos (3 * x) = 0)
have h_card_eq_4 : S.card = 4
```

Now, use the same process for the following theorem:

{**theorem**}

You must follow all the mandatory rules above. After deep consideration, provide the Lean 4 intermediate step code snippets. While ensuring correctness, the more intermediate steps, the better.

```
Prompt for Dynamic Replanning
```

You are an expert assistant specializing in Math Olympiads and the Lean 4 theorem prover, with a particular talent for proof decomposition and overcoming difficult proof steps.

Your primary goal is to refine an existing proof plan by inserting more granular, logically sound subgoals to help a prover overcome a specific, identified bottleneck.

It is crucial to strictly adhere to the following rules—any violation will be considered an error.

**Task**
Given a Lean 4 theorem, its initial proof plan, and a specific `have` statement where a prover has become stuck, your task is to generate a new, **complete proof plan**.

This new plan must include all the original steps, but with additional, simpler `have` statements inserted **immediately before** the 'stuck' subgoal. These new steps must logically lead to the proof of the stuck subgoal, breaking down the complex reasoning into a series of more manageable steps.

**Mandatory Rules**
You must comply with every rule in this section. Failure to adhere to any single rule will result in an incorrect output.

(The first 16 rules are identical to those in the `Prompt for Initial Planning` and must be strictly followed.) In addition, the following task-specific rules apply:

17. **Insert Before Stuck Step**: The new auxiliary `have` statements must be inserted **immediately before** the provided 'stuck' subgoal.

18. **Provide Complete Plan**: The output must be the **entire, updated plan**, including all original and new `have` statements in the correct order. Do not output only the new steps.

19. **Logical Progression**: The newly inserted steps must be logically sound and serve as direct prerequisites for proving the stuck subgoal. They should bridge the logical gap that caused the prover to get stuck.

**Below is an example to illustrate the process:**

**Theorem:**

```
theorem trig_identity_4x (x : ℝ) :
  Real.sin (4 * x) = 4 * Real.sin x * Real.cos x * (1 - 2 * Real.sin
    x ^ 2) := by
```

**Initial Plan:**

```
  have h_sin4x_is_2sin2xcos2x : Real.sin (4 * x) = 2 * Real.sin (2 *
    x) * Real.cos (2 * x)
  have h_final_identity : 2 * Real.sin (2 * x) * Real.cos (2 * x) = 4
    * Real.sin x * Real.cos x * (1 - 2 * Real.sin x ^ 2)
```

**Stuck Subgoal:**

```
  have h_final_identity : 2 * Real.sin (2 * x) * Real.cos (2 * x) = 4
    * Real.sin x * Real.cos x * (1 - 2 * Real.sin x ^ 2)
```

**Refined Plan (Your Output):**

```
  have h_sin4x_is_2sin2xcos2x : Real.sin (4 * x) = 2 * Real.sin (2 *
    x) * Real.cos (2 * x)
  have h_sin2x : Real.sin (2 * x) = 2 * Real.sin x * Real.cos x
```

```
have h_cos2x_in_terms_of_sin_cos : Real.cos (2 * x) = Real.cos x ^
   2 - Real.sin x ^ 2
have h_cos2x_in_terms_of_sin : Real.cos (2 * x) = 1 - 2 * Real.sin
   x ^ 2
have h_final_identity : 2 * Real.sin (2 * x) * Real.cos (2 * x) = 4
   * Real.sin x * Real.cos x * (1 - 2 * Real.sin x ^ 2)
```

Now, use the same process for the following three items:

{**theorem**}

{**initial_plan**}

{**stuck_subgoal**}

You must follow all the instructions and mandatory rules above. After deep consideration, provide the complete, refined Lean 4 plan.

