# OpenReview forum: "Scaling up Multi-Turn Off-Policy RL and Multi-Agent Tree Search for LLM Step-Provers"
_ICLR.cc/2026/Conference — Submitted to ICLR 2026_

### Official Review · Reviewer_be4i · 2025-10-20

**Soundness:** 3
**Presentation:** 2
**Contribution:** 3
**Rating:** 4
**Confidence:** 4

**Summary:**

The paper presents BFS-Prover-V2, a theorem proving system that uses multi-turn off-policy reinforcement learning and multi-agent tree search to scale both training and inference in LLM theorem proving. The model shows sota empirical results on existing benchmarks such as minif2f and proofnet, showing the strength of using BFS in LLM theorem proving.

**Strengths:**

This work's strongest motivation would lie in combining multi-turn off-policy RL with planner-guided inference for theorem proving. In specific, the planner-enhanced multi-agent search shows a promising way in theorem proving. The methodology is explained in a very clear and thorough way, making it easy to understand. Empirically, BFS-Prover-v2 achieves high performance, which showcases the strength of best first search.

**Weaknesses:**

1. One of the largest weaknesses would be the computation. Since it requires large-scale multi-turn and retraining cycles, it would be computationally expensive. The authors did not discuss the computation.

2. Result elaborations are insufficient, with limited interpretation of why specific design choices show observed gains.

3. The evaluations are only done on two datasets (minif2f and proofnet); other datasets such as putnambench, deepseek-prover-bench, and others would strengthen the generalisability.

4. The paper omits ablation studies, making it unclear how much each component (adaptive filtering, retraining, planner) contributes individually.

5. The evaluation metrics are not discussed – it's unclear whether the comparisons are measured in pass@n or other metrics.

**Questions:**

1. Could the author provide a deeper discussion on the results, such as the proof diversity and error cases and other aspects?

2. Computational efficiency would be a large issue since it requires multi-turn. What were the computational resources (GPUs, hours, and others) used for reproducibility?

3. Would the system maintain performance under reduced computational budgets or smaller-scale models?

4. Are the metrics comparable to other works pass@n evaluations?

---

> ### Author Response · Authors · 2025-11-21
>
> # [Part 1/2]
>
> We are deeply grateful for your thorough and constructive review. Your feedback has been instrumental in strengthening the empirical rigor of our work. We have submitted a revised version of the paper that strictly adheres to your valuable advice.
>
> Below, we address your concerns regarding computational resources, ablations, and evaluation metrics. (Please see the subsequent comment [Part 2/2] for our response to your specific Questions.)
>
> ## Response to Weaknesses
>
> ### W1 - Computational Cost
>
> We completely agree that computational efficiency is crucial and the initial submission lacked a thorough discussion on this. We have now included a detailed breakdown of our training and inference computational budget in the revision.
> - **Search Compute in Expert Iteration**: As explicitly detailed in **Section 2.2 (Phase 1, top of Page 4)**, each round of expert iteration involves **∼10^7 tree searches**.
> - **Training Budget**: In the newly added **Section 3.1 (Training Setup)**, we clarified the training budget: a standard expert iteration requires only **1 epoch**, whereas retraining from the base checkpoint requires **3 epochs**. The computational cost per epoch is standard full-parameter SFT for a 7B/32B model.
> - **Data Analysis**: We also included the amount of training data used when analyzing the **Periodic Retraining** in the new **Section 3.3**.
> - **Inference Budget**: Following your suggestion, we have added a comprehensive inference computational budget analysis in the new **Section 3.4** in our revision, providing detailed breakdowns of resource requirements for reproducibility.
>
> ### W2 - Result Elaborations
> Thank you for highlighting this critical gap. We have now added extensive ablation studies and interpretations in our revision. These include:
> - Comprehensive ablations for all training and inference components (**Section 3.3-3.4**).
> - More detailed analysis explaining why each design choice yields observed gains, including specific **state-action pair examples** illustrating the advantages of our data curation method.
>
> ### W3 - Dataset Generalizability
> We acknowledge your suggestion for broader benchmarking. We selected miniF2F and ProofNet to strictly evaluate two distinct capabilities:
> - miniF2F: Represents In-Distribution performance (Olympiad-level problems similar to training data).
> - ProofNet: Represents Out-Of-Distribution (OOD) generalization (Undergraduate-level mathematics). We believe this combination robustly assesses the prover's adaptability. Our strong OOD results on ProofNet (exceeding baselines by a large margin) provide evidence of true reasoning generalization rather than overfitting to a specific dataset. A more detailed analysis is provided in **Section 3.2 (Benchmark Results)**.
>
> ### W4 - Ablation Studies
> Following your advice, we have now added comprehensive ablation studies in **Section 3.3 and 3.4**.
> - **Components Tested**: Ablations cover **tactic filtering**, **periodic retraining**, **model scaling**, **planner agents**, and **subgoal caching**.
> - **Quantification**: Each component's individual contribution to the final success rate is now clearly quantified.
>
> ### W5 - Evaluation Metrics
> We have now clearly specified our evaluation protocol in **Section 3.4 (Computational Budget)** regarding the metric. We report accumulative performance to ensure academic rigor and to truthfully reflect the model's maximum reasoning capability. This metric aggregates results across a small grid of hyperparameter configurations. Also, as we detail in the **response to Q4** below, our metric (pass@8192 instances) is designed to be comparable to standard pass@N metrics in terms of total computational budget.
>
> **Please see the subsequent comment [Part 2/2] for our response to your specific Questions.**

---

> > ### Author Response · Authors · 2025-11-21
> >
> > # [Part 2/2]
> >
> > ## Response to Questions
> >
> > ### Q1 - Discussion on Results
> > We have substantially expanded our analysis in our revised manuscript:
> > - **Proof Diversity**: **Appendix A** includes detailed case studies showing how step-level proving discovers novel, more concise proofs compared to human or whole-proof approaches.
> > - **Concrete Examples**: We added state-action pairs demonstrating our tactic filtering method's advantages in Section 2.2.
> > - **Planner Effectiveness**: **Appendix B** provides comprehensive case studies of how the planner-prover paradigm effectively navigates complex search spaces.
> >
> > ### Q2 - Computational Resources
> > We provide complete computational details for reproducibility. Please refer to our detailed breakdown in **Response to W1**.
> >
> > ### Q3 - Performance Under Reduced Budgets
> > We have thoroughly investigated and explicitly analyzed the system's performance with smaller models and reduced budgets in the revision:
> > - **7B Model Results**: We added experiments with a **7B model**. On ProofNet, it achieves **34.4%** (comparable with much larger models). The **7B + Planner** combination on miniF2F reaches **92.6%**, achieving SOTA-level performance.
> > - **Training Dynamics**: The new **Figure 4** shows performance evolution across training checkpoints, demonstrating strong performance **even with fewer training iterations**.
> > - **Conclusion**: This demonstrates that our approach remains highly effective even with smaller models and reduced computational budgets.
> >
> > ### Q4 - Metric Comparability
> > We have clarified this important point in the new **Section 3.4 (Computational Budget)**: While tree-search and whole-proof models operate differently, our evaluation ensures a fair comparison based on total instance budget:
> > - **Instance Lifecycle**: We define `pass@8192` as cumulatively launching 8,192 distinct prover instances. Although each instance has a maximum lifecycle of 600 seconds, the vast majority terminate significantly earlier because they either successfully find a proof or hit a dead end (empty search tree).
> > - **Comparability**: Therefore, our `pass@8192` represents a budget of proof attempts comparable to the `pass@N` used in whole-proof generation models.
> >
> > ## Conclusion
> > We are genuinely grateful for your thoughtful review and the opportunity to improve our work. Your feedback has led to substantial improvements in our paper's clarity, completeness, and scientific rigor. We hope our revisions address your concerns satisfactorily.
> > We would be deeply honored if you would consider raising your score. Thank you again for your time and expertise.

---

> ### Author Response · Authors · 2025-11-26
>
> We are writing to kindly remind you that we posted our response 5 days ago. If you have any additional feedback, concerns, or questions regarding our response, we would greatly appreciate hearing from you.

---

> > ### Comment · Reviewer_be4i · 2025-11-27
> >
> > Thank you for the additional clarifications. However there remain several major limitations that raise further concerns:
> >
> > 1. Computational cost remains a substantial issue. Although the computational cost is mentioned, it is still unclear how many, or what GPUs, and time were used and what the full computational setup was. The large multi-turn setting requires significant computational resources, which poses a challenge for reproducibility.
> >
> > 2. Evaluating only on ProofNet and MiniF2F would be insufficient to demonstrate generalisability. Additional datasets were recommended in the initial round of feedback, but these have not been tested.
> >
> > 3. MiniF2F alone is insufficient, as it is well known to be potentially contaminated (as we can see the accuracies are really high). Evaluations on other datasets reported in Figure 4, such as PutnamBench, where performance is lower, would be valuable to include.
> >
> > 4. The use of pass@8192 is extremely computationally expensive, which further exacerbates the concerns raised in point (1).

---

> ### Author Response · Authors · 2025-12-03
>
> We are very grateful for your careful follow up and for taking the time to re examine our work. Below we respond point by point.
>
> ## 1. Computational Cost
>
> As you pointed out, the computational budget is already described in the revised manuscript and in our previous response, but we agree that it is helpful to make this even more explicit. To be precise, standard full-parameter SFT training for our 7B model requires approximately 250-300 A100 GPU hours per epoch, and the 32B model requires approximately 1,500-1,800 A100 GPU hours per epoch. These are standard costs for models of this size and can be converted to other computational setups.
>
> We also clarify how we view compute in the context of this work: One of the main goals and contributions of our paper is to study a **scaling methodology** for search based theorem proving: how performance evolves as both model size and training/inference compute are increased. Small scale experiments can appear promising after a few epochs, but often do not reveal whether the method truly continues to improve or saturates at larger scales. From this perspective, the substantial compute is **part of the scientific question itself we are studying**, and is **intended to demonstrate the robustness and scalability of our approach at the frontier of performance**, rather than being a drawback.
>
> As noted in our revision (and Figure 4), the specific bottlenecks and breakthrough phenomena we describe are observable and reproducible even with smaller models and fewer epochs, ensuring that our scientific insights remain accessible under reduced budgets to researchers with more limited resources.
>
> ## 2. Additional evaluations
>
> We have conducted additional evaluations on DeepSeek-ProverBench.
>
> As shown in the table below, our BFS-Prover-V2-32B (w/ planner) achieves a score of 3.08, significantly outperforming strong baselines.
>
> | Model                               | DeepSeek-ProverBench |
> |------------------------------------|------------------------------|
> | DeepSeek-Prover-V2-7B              | 0.31                         |
> | Kimina-Prover-Distill-8B           | 1.38                         |
> | Kimina-Prover-72B                  | 1.53                         |
> | Goedel-Prover-V2-8B                | 1.53                         |
> | Goedel-Prover-V2-32B               | 1.85                         |
> | Goedel-Prover-V2-32B (w revision)  | 2.46                         |
> | **BFS-Prover-V2-32B**              | **1.85**                     |
> | **BFS-Prover-V2-32B (w planner)**  | **3.08**                     |
>
> Note: Baseline data partially sourced from [Goedel-Prover-V2 Openreview official comments](https://openreview.net/forum?id=j4C0nALrgK&noteId=2JXmBmmG2H).
>
>
> ## 3. Validity of MiniF2F
>
> We understand your concern regarding potential dataset contamination, given the high scores. However, we respectfully offer the following context:
>
> **Community Standard**: MiniF2F remains the primary benchmark for formal theorem proving. Until last year, the SOTA was below 50%. When we started this project, the best known score was 72.95 percent from BFS-Prover-V1 earlier this year. We believe that the recent jump is better explained by rapid recent progress of the community rather than by leakage.
>
> **Timeline**: MiniF2F was released in 2021. If leakage were the primary driver of high scores, we would have expected saturation much earlier. The fact that saturation is only happening now, in late 2025, validates the difficulty of the benchmark and reflects the community's genuine progress in neural theorem proving.
>
> **Strict Decontamination**: We applied rigorous filtering to our training data. The formal solutions for miniF2F problems do not exist in any training data source we utilized.
>
> In addition, we believe the strong gains on out of distribution benchmarks such as ProofNet and DeepSeek-ProverBench further address the concern about miniF2F.
>
> ## 4. Justification for Pass@8192
>
> The main reason we adopt pass@8192 is comparability with existing work. Our primary baselines, such as Kimina Prover 72B and DeepSeek Prover 671B, as well as the concurrent Goedel Prover V2 32B, all utilize the Pass@8192 and accumulative metric. To ensure that comparisons are fair under a similar total number of proof attempts, we therefore evaluate our 7B and 32B models under the same pass@8192 budget. Since our models are much smaller, the total compute required to reach pass@8192 are lower than the larger baseline models we are comparing against.

---

### Official Review · Reviewer_MFzr · 2025-11-01

**Soundness:** 3
**Presentation:** 3
**Contribution:** 3
**Rating:** 6
**Confidence:** 5

**Summary:**

This paper proposes BFS-Prover-V2, a step-level automated theorem proving system using LLMs. This system addresses two major challenges in LLM-based provers: training-time scaling (performance plateaus) and inference-time scaling (search complexity).

At training time, it introduces a multi-stage expert iteration inspired by AlphaZero. This includes two novel learning techniques to overcome prover performance plateaus: "adaptive tactic filtering" and "periodic retraining."

At inference time, it employs a "planner-enhanced multi-agent architecture." This is a hierarchical approach where a high-level "planner" agent (a general-purpose LLM) decomposes complex theorems into simple subgoals, and a team of specially trained "prover" agents solves them in parallel.

As a result of experiments, BFS-Prover-V2 achieved state-of-the-art (SOTA) performance as a step-level prover on the miniF2F (95.08%) and ProofNet (41.4%) benchmarks.

**Strengths:**

1.  **Achieving new SOTA in the Step-Level Approach:**
    This research is significant for achieving clear state-of-the-art (SOTA) performance in the "step-level generation" approach, which is distinct from "whole proof generation" (the one-shot generation of an entire proof).
2.  **Uniqueness and Future Potential of the Approach:**
    The step-level approach may possess distinct advantages over whole proof. Specifically, it could play a crucial role in exploring unknown theorems, particularly through interactive collaboration with proof assistants like Lean or by dynamically changing strategies mid-proof. The AI for Math community needs to continue supporting and evaluating this different approach, and this paper is a strong contribution in that direction.
3.  **Robust Training and Inference Architecture:**
    The "adaptive filtering" and "periodic retraining" introduced to overcome performance plateaus during training are robust engineering solutions for practical issues in long-term RL training. Additionally, the hierarchical decomposition by the planner at inference time, combined with "dynamic replanning" when the prover gets stuck, represents a very powerful paradigm for addressing complex problems.

**Weaknesses:**

1.  **Limitation of the Expert Iteration Design (Absence of the Planner):**
    The proposed expert iteration (training loop) focuses solely on enhancing the prover's (tactic executor) capability, and lacks a mechanism to strengthen the planner (strategy planner).
2.  **Potential Scaling Bottleneck:**
    In the current architecture, the planner's performance (i.e., its problem decomposition ability) relies on the fixed capability of an external, general-purpose model (Gemini 2.5-pro). However powerful the prover becomes, the planner's performance could become the ultimate bottleneck for the entire system, potentially causing performance to plateau.
3.  **Limited Comparative Benchmark Scores:**
    While SOTA for a step-level prover, the absolute score on miniF2F (95.1%) falls short of the leading whole proof approaches, such as Seed-Prover (99.6%) and Delta-Prover (95.9%).

**Questions:**

A primary weakness of this paper is that the planner is not strengthened within the expert iteration loop. I would like to ask the authors' views on a more integrated expert iteration design that strengthens the planner in parallel with the prover's performance improvements—for example, by training the planner using success/failure feedback from the prover.

---

> ### Author Response · Authors · 2025-11-21
>
> We are truly honored by your encouraging review and your recognition of our work. We deeply value your insight that the AI-for-Math community needs to continue supporting the step-level research line, and we are grateful for the opportunity to clarify our design choices. We address your specific concerns regarding benchmark comparisons and the planner design below.
>
> ## Response to W1, W2, Q1: Planner Design
>
> We deeply appreciate your visionary suggestion to train the Planner within the expert iteration loop. While this is indeed our roadmap for future work, we deliberately chose to use a general-purpose LLM for the current version based on the following scientific rationale:
>
> - **Efficiency of General Intelligence**: At the current stage, state-of-the-art general-purpose LLMs possess strong reasoning, planning and decomposition capabilities derived from massive pre-training and post-training. Leveraging these "off-the-shelf" reasoning capabilities is much more efficient and yields better immediate performance than training a specialized, smaller planner from scratch without a mature dataset of high-quality decompositions.
>
> - **The "Data Engine" Strategy**: We view BFS-Prover-V2 as a foundational data engine. By using a strong general model to guide the search, we are actively collecting successful (Theorem → Decomposition → Proof) trajectories. This high-quality synthetic data is precisely what is needed to fine-tune a specialized, smaller planner for future work.
>
> - **Focus on Prover Scaling**: Our primary focus in this work was to validate the dual-scaling strategy (training-time and inference-time). Fixing the planner allowed us to isolate and rigorously verify the training-time scaling strategies of the Prover model without the confounding factors that would arise from simultaneously training two interdependent models.
>
> ## Response to W3: Benchmark Scores Comparison
>
> We acknowledge that on miniF2F, our score (95.1%) is marginally lower than recent whole-proof models like Seed-Prover (99.6%) and Delta-Prover (95.9%). We respectfully offer three perspectives on this:
>
> - **Performance Saturation**: Both 95.1% and 99.6% represent performance near the saturation point of the miniF2F benchmark. At this level, the marginal difference is less effective at differentiating formal reasoning capabilities. This is why we also emphasize our **OOD performance on ProofNet** (41.4%), where our step-level approach significantly outperforms much larger models like DeepSeek-Prover-V2-671B (37.1%).
> - **ICLR Policy on Contemporaneous Work**: Per the **[ICLR 2026 Reviewer Guide (last Q&A)](https://iclr.cc/Conferences/2026/ReviewerGuide)**, authors are not required to compare with papers published (in peer-reviewed venues) on or after July 24, 2025, nor are they required to compare with papers solely on arXiv. Both Seed-Prover and Delta-Prover are arXiv preprints from July 2025. While we included these comparisons for the sake of scientific completeness and transparency, strictly speaking, the performance gap against these contemporaneous arXiv works should not constitute a weakness in the assessment of our work's contribution in this submission cycle.
> - **Unique Value**: As you insightfully pointed out, the primary contribution of this work is pushing the boundary of the Step-Level/Tree-Search paradigm. This approach offers distinct advantages in interactivity and process visibility that whole-proof models cannot provide, which we believe holds independent value regardless of marginal score differences on saturated benchmarks.
>
> ## Conclusion
>
> We are deeply grateful for your constructive feedback and your acknowledgment of our work's quality. We have made every effort to engage with your points by detailing our perspectives. Given that our step-level approach offers a unique and promising direction for the community which is distinct from the saturated whole-proof paradigm, we respectfully hope that you consider raising your score to champion the acceptance of this work.

---

> ### Author Response · Authors · 2025-11-26
>
> We are writing to kindly remind you that we posted our response 5 days ago. If you have any additional feedback, concerns, or questions regarding our response, we would greatly appreciate hearing from you.

---

### Official Review · Reviewer_mxxM · 2025-11-04

**Soundness:** 2
**Presentation:** 2
**Contribution:** 3
**Rating:** 4
**Confidence:** 4

**Summary:**

In this paper, they proposed a step-level theorem proving system of BFS-Prover-V2 by improving both training-time and test-time scaling. In training-time scaling, they introduced a multi-turn scaling-up strategy of repeatedly expert iteration, data synthesis and retrain the model from the checkpoint. In test-time scaling, they introduced both planner and prover agents: planner as a general-purpose reasoning LLM for task decomposition and prover as an LLM-based tactic generator and multi-agentic system for enhancing reasoning capability in the inference time.

In experiments, it is regrettable that the authors compared their model with a limited number of models in the main result of Table 1. In a first look of the Table 1, the proposed BFS-Prover-V2-32B is best among the step-level prover, while some other models like DeltaProver exploit the lemma decomposition strategy and it can be classified as “step-level” models although it is placed in “Whole-proof provers.” Similarly, some results from the latest models are missing from Table 1 such as Baba et al 2025 and Goedel-Prover-V2. Considering this incompleteness of the main result, I am hesitant to adequately assess the effectiveness of the proposed methodology.

**Strengths:**

- S1: Interesting improvement in training-time scaling and test-time scaling
- S2: Figure 2 is very interesting: it presents the real training-time performance improvement including the model scaling up.

**Weaknesses:**

- W1: The overall work seems still on-going: if either training-time scaling or test-time scaling is effective, it can be evaluated via multiple ablation studies. In this work, the effectiveness of the Planner is suggested with the BFS-Prover-V2-32B case while it is not presented within the 7B model. Indeed, Table 1 is the unique final result table in this paper and it is difficult to evaluate the effectiveness of this model only from this table.
- W2: The main table 1 is indeed limited and the statement “Our system sets a new state of the art for LLM step-provers” doesn’t sound plausible. DeltaProver is classified as “Whole-proof provers” and some import results are missing.
- W3: The overall writing can be improved. Before I reach the final results at the end of the paper, I see many intermediate results that many readers would like to see after the main result. This may prevent readers from grasping the overall contribution of this paper.

**Questions:**

- Q1: Do you have some specific reason why you have classified DeltaProver as whole-proof provers?
- Q2: It seems that you have scaled up the model from 7B to 32B at the 13th expert iteration. Do you have some clarification or specific reason for this?

---

> ### Author Response · Authors · 2025-11-21
>
> We are deeply grateful for your thorough and constructive review. We have submitted a revision that incorporates your valuable suggestions. We address your specific concerns point by point below.
>
> ## Response to Summary
>
> Thank you for your careful attention to our comparison table. We would like to respectfully clarify two important points:
> 1. Goedel-Prover-V2: We clarify that the model listed as "Goedel-Prover-32B" in Table 1 in our original submission is indeed the Goedel-Prover-V2-32B model. To improve clarity based on your feedback, we have renamed it to Goedel-Prover-V2-32B in the revision.
> 2. Baba et al. 2025: While we cited Baba et al. 2025 in our introduction section, we have now also added their results to Table 1 in the revised version. We believe our main results table now includes all representative theorem proving models available at the time of submission.
>
> ## Regarding Delta-Prover Classification (Q1)
>
> We deeply appreciate your question about Delta-Prover's classification, which highlights an important distinction in formal theorem proving approaches.
>
> The conventional naming in this research line distinguishes "step-level provers" as models that perform tactic-level interaction with the Lean environment, generating one tactic at a time and receiving feedback before proceeding (e.g., HyperTree Proof Search, ReProver, AlphaProof, InternLM-StepProver, BFS-Prover-V1).
>
> While we fully agree that DeltaProver is an impressive work (which we cite and compare against), its fundamental mechanism differs. It performs top-down sketch decomposition and generates complete proof scripts for subgoals rather than interactive tactic-level generation. This distinction reflects a broad community consensus. Similar definitions and classifications are found in MPS-Prover[1] (p.6, top table), Hunyuan-Prover[2] (p.6, top table), DSP+[3] (p.4, bottom definition), Seed-Prover[4] (p.2, 2nd paragraph), and BFS-Prover-V1[5] (p.8, top table).
>
> However, we fully accept your feedback that the term "Step-Level" can be ambiguous in a broader context. To improve clarity, we have adopted the terminology used by Kimina-prover[6] and DeepSeek-Prover-V2[7], renaming our category from "Step-Level Provers" to "Tree-Search Provers" in our revised table. We hope this accurately reflects the methodological differences while respecting the capabilities of models like DeltaProver.
>
> ## Response to Weaknesses
>
> ### W1
> - Ablation Studies: We completely agree that comprehensive ablations are crucial. In response, we have:
> - Added dedicated "Further Analysis" sections with extensive ablations on both training and inference components
> - Included Planner+7B model results in Table 1 to demonstrate effectiveness across model sizes
> - Provided detailed analysis of each component's contribution
>
> ### W2
> - Main Table Limitations: As addressed above, we have updated Table 1 with missing results and clarified our categorization. We humbly acknowledge that our initial presentation could have been clearer.
>
> ### W3
> - Writing and Organization: Thank you for this constructive feedback. We have thoroughly revised the paper structure based on your advice:
> - Modularization: We have reorganized the manuscript to clearly separate the proposed methodology from the experiments. Section 2 now focuses solely on the design of our training and inference pipeline.
> - Consolidated Experiments: All experimental results, including the intermediate ablations and the "Further Analysis" (covering training/inference ablations), are now unified in the Experiment section. Intermediate results now appear after the main results for better flow. This allows readers to access the main contributions (Table 1) earlier, followed by the in-depth analysis.
> - Polishing: We have thoroughly proofread the entire paper to improve academic rigor and flow.
>
> ## Response to Q2 - Model Scaling Timing
>
> Thank you for this keen observation. As shown in Figure 4 in the revised paper, around checkpoint 16, we observed the 7B model approaching capacity limits with diminishing returns from expert iteration. We strategically scaled to 32B to:
> 1. Continue the iteration process with greater model capacity
> 2. Validate the scalability of our multi-stage expert iteration approach
> The results confirm our approach's effectiveness across model sizes. Detailed analysis is now provided in Section 3.3 under Periodic Retraining and Model Scaling.
>
> ## Conclusion
> We are deeply grateful for your thoughtful review and hope our revisions adequately address your concerns. We believe the improvements made in response to your feedback have substantially strengthened the paper. We remain committed to further discussion if you have any remaining questions and would like to incorporate any additional suggestions you may have.
> Given that the main grounds for the initial assessment have been addressed with our clarifications and revision, we respectfully hope that you consider raising your score to reflect these improvements.

---

> > ### Author Response · Authors · 2025-11-21
> >
> > **Reference**
> >
> > [1] [MPS-Prover](https://arxiv.org/abs/2505.10962)
> >
> > [2] [Hunyuan-Prover](https://arxiv.org/abs/2412.20735)
> >
> > [3] [DSP+](https://arxiv.org/abs/2506.11487)
> >
> > [4] [Seed-Prover](https://arxiv.org/abs/2507.23726)
> >
> > [5] [BFS-Prover-V1](https://arxiv.org/abs/2502.03438)
> >
> > [6] [Kimina-Prover](https://arxiv.org/abs/2504.11354)
> >
> > [7] [DeepSeek-Prover-V2](https://arxiv.org/abs/2504.21801)

---

> ### Author Response · Authors · 2025-11-26
>
> We are writing to kindly remind you that we posted our response 5 days ago. If you have any additional feedback, concerns, or questions regarding our response, we would greatly appreciate hearing from you.

---

### Author Response · Authors · 2025-11-21
**Updated paper**

Dear reviewers,

We sincerely thank you for your time and the constructive feedback provided during the review process. Your insights have been invaluable in helping us strengthen the empirical rigor and clarity of our work.

We have uploaded a **revised manuscript** that incorporates your suggestions. We would be grateful if you could take a moment to look over the changes. Key updates include:

- **Extensive Ablation Studies**: We have added comprehensive ablations for both training and inference components (e.g., tactic filtering, periodic retraining, planner integration, subgoal proof caching) in the new **Sections 3.3 and 3.4**, explicitly quantifying the contribution of each module.

- **Expanded 7B Model Experiments**: We have included additional experiments with the **7B model** to demonstrate the scalability and effectiveness of our approach on smaller models and reduced budget.

- **Deepened Analysis & Case Studies**: We have provided a more thorough analysis of why our methods work. This includes adding specific **state-action pair examples** to illustrate the mechanism of adaptive tactic filtering , as well as tracking **training dynamics** (Figure 4)  to demonstrate the efficacy of periodic retraining.

- **Computational Transparency**: We have added detailed breakdowns of our training and inference computational budgets in the new **Sections 3.1, 3.3, and 3.4**, ensuring full reproducibility .

- **Metric Clarification**: We have clarified our evaluation protocol to ensure fair comparisons with standard whole-proof metrics.

We have also posted detailed individual responses to each reviewer addressing their specific concerns and questions. We remain available for any further discussions and respectfully hope that the revisions address your concerns.

Sincerely,

The Authors

---

### Author Response · Authors · 2025-12-03
**Final Rebuttal Summary for BFS-Prover-V2**

Dear reviewers and AC,

We sincerely thank the reviewers and the area chair for their time, patience, and thoughtful engagement with our submission, especially under the unusually heavy workload created by this year’s rollback process. We deeply appreciate the careful attention given to our work.

In addition to the **update summary** posted earlier in the discussion, we have made a number of further improvements in direct response to the reviewers’ follow-up comments. These include **new evaluations**, **expanded analyses**, and **clarifications** beyond what was originally outlined. Specifically, we added the new DeepSeek-ProverBench results, clarified the computational cost with explicit GPU hour estimates for both model sizes, provided additional explanations on pass budget comparability, and strengthened the discussion on dataset validity and benchmark saturation. These updates are complementary to the earlier revisions, which included **comprehensive ablation studies**, **expanded experiments**, **deeper analysis of training dynamics**, **additional case studies** illustrating why our techniques work, and clearer descriptions of our evaluation metrics and computational setup.

We are grateful for the constructive feedback that guided these improvements. The review process has helped us further strengthen the clarity, completeness, and scientific rigor of our paper. Thank you again to the reviewers and the AC for your effort during this atypical review cycle. We hope the revisions make the contribution of our work clearer and more compelling.

Sincerenly,

The Authors

---

### Meta-Review · Area_Chair_XBty · 2025-12-19

**Summary:**

This paper proposes to use multi-turn off-policy RL to improve theorem proving. While the paper has some interesting points, there were several points that perhaps make it not yet ready for publication. The most central one is the computational cost of the method, with benefit which is not clearly proportional to such use. I appreciate that the authors in a  revision made the computational cost most transparent, but I couldn't quite f figure out from their text whether the extra computational cost is worth the improvements in performance  to begin with. That is not clearly discussed.  What can we automatically prove with their model that we couldn't before?

**Reviewer Concerns:**

several costs, including computational cost.

**Reviewer Scores:**

I cannot predict that.

---

### Decision · Program_Chairs · 2026-01-26

Reject